# High-Performance Arithmetic Circuit Optimization via Differentiable Architecture Search

**Xilin Xia**[1] *   **Jie Wang**[1] †   **Wanbo Zhang**[1]   **Zhihai Wang**[1]
**Mingxuan Yuan**[2]   **Jianye Hao**[2,3]   **Feng Wu**[1]

[1] MoE Key Laboratory of Brain-inspired Intelligent Perception and Cognition,
University of Science and Technology of China
[2] Noahs Ark Lab, Huawei Technologies
[3] College of Intelligence and Computing, Tianjin University

## Abstract

Arithmetic circuit optimization remains a fundamental challenge in modern integrated circuit design. Recent advances have cast this problem within the Learning to Optimize (L2O) paradigm, where intelligent agents autonomously explore high-performance design spaces with encouraging results. However, existing approaches predominantly target coarse-grained architectural configurations, while the crucial interconnect optimization stage is often relegated to oversimplified proxy models or a heuristic approach. This disconnect undermines design quality, leading to suboptimal solutions in the circuit topology search space. To bridge this gap, we present ARITH-DAS, a **D**ifferentiable **A**rchitecture **S**earch framework for **Arith**metic circuits. To the best of our knowledge, ARITH-DAS is the first to formulate interconnect optimization within arithmetic circuits as a differentiable edge prediction problem over a multi-relational directed acyclic graph, enabling fine-grained, proxy-free optimization at the interconnection level. We evaluate ARITH-DAS on a suite of representative arithmetic circuits, including multipliers and multiply-accumulate units. Experiments show substantial improvements over state-of-the-art L2O and conventional methods, achieving up to **27.05%** gain in hypervolume of area-delay Pareto frontiers, a standard metric for evaluating multi-objective optimization performance. Moreover, integrating our optimized arithmetic units into large-scale AI accelerators yields up to **6.59%** delay reduction, demonstrating both scalability and real-world applicability.

## 1   Introduction

Arithmetic circuits, including multipliers and multiply-accumulate units, constitute the computational foundation of modern hardware platforms such as CPUs, GPUs, AI accelerators, and digital signal processors [1–3]. These circuits perform essential arithmetic operations that dominate the computational workload across a broad range of compute-intensive applications [4–6]. In deep neural networks, for example, multiplication operations account for over 99% of total computation [7]. As machine learning and high-performance computing systems continue to grow in scale and complexity, optimizing arithmetic circuits for both latency and area efficiency has become a critical challenge for enabling scalable and effective AI hardware systems.

Arithmetic circuit optimization poses a fundamental NP-hard challenge in discrete combinatorial optimization. The discrete design space scales exponentially with the bit width, reaching a complexity of $O(2^{4N^2})$ as detailed in Appendix B. Recent studies have reformulated arithmetic circuit

---

*This work was done when Xilin Xia was an intern at Huawei Noahs Ark Lab.

†Correspondence author. Email: <jiewangx@ustc.edu.cn>

The code is available at `https://github.com/MIRALab-USTC/Arith-DAS.git`

39th Conference on Neural Information Processing Systems (NeurIPS 2025).

optimization as a Learning to Optimize (L2O) problem [3, 7–9], wherein learning-based agents are employed to explore efficient design strategies for performance improvement. These approaches typically initialize from expert-crafted designs, iteratively refine local structures, and leverage performance improvements between successive designs as reward signals to guide the optimization process, achieving promising results.

However, despite the notable successes, existing L2O approaches remain predominantly confined to coarse-grained architectural optimization, which primarily focuses on basic component allocation at each bit position. Fine-grained interconnect routing is still governed by heuristic rules that overlook the complex structural constraints of the design space, thereby compromising circuit quality. Recent efforts [10, 11] have attempted to model the interconnection in the arithmetic circuits through permutation matrix generation, leveraging mixed-integer programming (MIP) and differentiable optimization to enhance interconnect assignment. However, these methods are fundamentally constrained by oversimplified proxy formulations, which fail to faithfully reflect complex post-synthesis physical metrics. As circuit complexity increases, the misalignment between proxy and actual performance escalates, causing the optimization to converge prematurely to inferior design solutions.

To address these challenges, we present ARITH-DAS , a differentiable architecture search framework that directly targets the interconnection design space in arithmetic circuits. In contrast to prior work focused on coarse architectural component allocation, ARITH-DAS formulates fine-grained interconnection optimization as an edge prediction task over a multi-relational directed acyclic graph, capturing signal-level structural semantics. It employs multi-relational graph neural networks with attention mechanisms to model connection probabilities across relation types, and adopts a proxy-free objective aligned with post-synthesis metrics to ensure fidelity. ARITH-DAS further integrates with high-level allocation optimizers, enabling unified, multi-granularity circuit optimization.

We conduct systematic evaluations on a suite of representative arithmetic circuits, including multipliers and multiply-accumulate (MAC) units. Experimental results show that our method consistently achieves Pareto dominance over state-of-the-art approaches in area and delay, with hypervolume improvements of up to **27.05**%. Furthermore, when integrating the optimized circuits into large-scale AI computing systems, our approach delivers up to **6.59**% latency reduction compared to the state-of-the-art baseline designs, demonstrating its scalability and practical engineering applicability.

Contributions of this paper are summarized as follows: **(1)** We identify key limitations in existing arithmetic circuit optimization approaches and reveal the critical role of the interconnection design space in determining post-synthesis performance. **(2)** To the best of our knowledge, ARITH-DAS is the first to formulate the fine-grained interconnect synthesis of arithmetic circuits as a differentiable architecture search problem. **(3)** We propose a novel differentiable architecture search framework tailored for arithmetic circuits, which combines multi-relational graph neural networks with Graphormer-style attention mechanisms. The entire framework is trained in a proxy-free manner, directly aligning with post-synthesis performance metrics. **(4)** Extensive experimental results across representative circuits demonstrate that our method achieves state-of-the-art post-synthesis performance, while maintaining strong applicability to real-world design flows.

## 2 Preliminary: Arithmetic Circuit Optimization

Arithmetic circuits are typically composed of three main components: the **Partial Product Generator (PPG)**, the **Compressor Tree (CT)**, and the **Carry Propagate Adder (CPA)**. As shown in Figure 1, PPG encodes the inputs into a matrix of partial products based on the arithmetic operations, using schemes such as AND-gate encoding, Booth encoding, or more complicated methods [12–14]. Then the compressor tree reduces this matrix to two rows through parallel compression, and finally the CPA adds them to produce the final output. The compressor tree dominates both area and delay of the entire arithmetic circuit, making it the primary target for arithmetic circuit optimization [3, 7, 15]. The optimization of compressor trees typically involves three key steps:

**Compressor Allocation** This step determines the number and types of compressors used at each column of the compressor tree. The allocation must satisfy basic constraints such as input/output balance and convergence to two or fewer final rows. It forms the architectural backbone of the compression process and directly influences both the area and delay of the overall circuit.

**Compressor Stage Assignment** Given a valid compressor allocation, the stage assignment step specifies the execution order of each compressor, subject to data dependency and topological con-

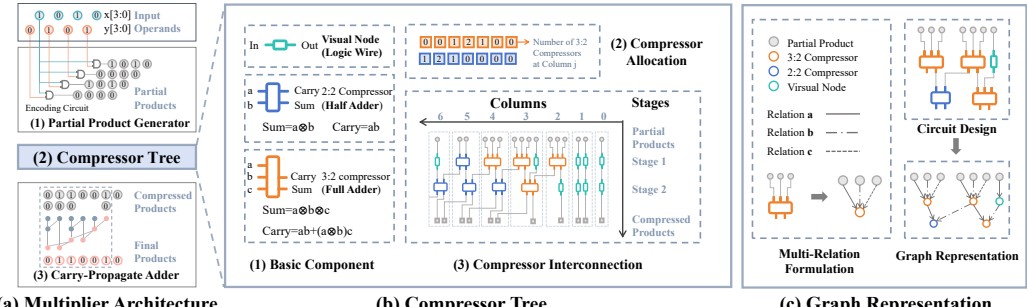

**(a) Multiplier Architecture**    **(b) Compressor Tree**    **(c) Graph Representation**

Figure 1: An illustration of a 4-bit multiplier with AND-gate based PPG.

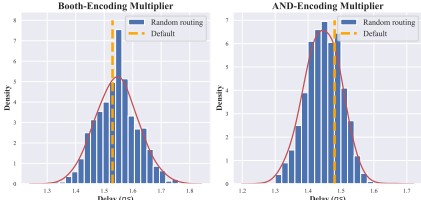

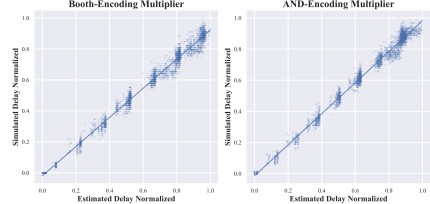

Figure 2: Delay distribution of 1000 randomly interconnected 16-bit multipliers using Wallace Tree compression with both AND and Booth partial product encoding PPG.

Figure 3: Comparison of estimated and post-synthesis delays for multipliers ranging from 4-bit to 64-bit, using Wallace Tree with both AND and Booth partial product encoding PPG.

straints. This process enforces legal signal propagation through the stages but does not yet define the exact ordering of partial product summation within each column. The resulting assignment determines the temporal alignment of computation without resolving interconnect details.

**Compressor Interconnection**  With allocation and stage assignment fixed, the final step involves establishing the logical interconnection between compressors, determining how outputs are routed to inputs across stages. This step has been largely overlooked in existing learning-to-optimize (L2O) frameworks, which typically assume simplistic sequential interconnection. Unlike allocation and stage assignment, which operate on regularized representations, interconnect optimization requires reasoning over graph-structured data and exposes a vast and irregular search space, making it both critical and challenging to optimize effectively.

## 3  Motivation Challenge: Interconnection Optimization

Existing arithmetic circuit optimization frameworks predominantly focus on allocation-level design using regularized structural representations, whereas interconnections between compressors are typically handled through simplistic sequential wiring schemes. However, recent studies reveal that interconnects constitute a critical and nontrivial design space [10, 11], with considerable impact on post-synthesis delay. In essence, existing approaches exhibit three fundamental limitations: **(1)** the neglect of interconnect-level optimization; **(2)** dependence on oversimplified heuristics and proxy delay models; **(3)** the absence of a unified framework for jointly optimizing coarse-grained allocation and fine-grained interconnect design. We elaborate on these limitations in the following discussion.

**Design Space of Interconnection**    The interconnection of compressors plays a critical role in determining the critical path of the compression tree. We illustrate this often-overlooked design space through a randomized experiment. Specifically, we perform random interconnection assignments on 16-bit multipliers using both AND-gate and Booth PPG schemes with fixed compressor allocation, and evaluate the post-synthesis delay of these randomly generated designs. The results are compared against the sequential interconnection strategy adopted by the recent state-of-the-art L2O method MUTE [7]. We present the post-synthesis delay distribution of the randomly routed designs in Figure 2, revealing that with a fixed compressor tree allocation, there exists a substantial timing optimization space of up to over $10\%$. Notably, the sequential wiring scheme does not exhibit a clear advantage over these randomized interconnection designs.

**Misaligned Proxy Objective**    Existing interconnect optimization methods leverage mixed-integer programming (MIP) [10] or differentiable approaches [11], both relying on oversimplified proxy

delay models. However, due to the complex and nonlinear nature of post-synthesis delays, which are closely coupled with the technology library and EDA toolchain, such proxy objectives often fail to reflect the true performance characteristics and lead to suboptimal routing solutions. To illustrate this, we conduct randomized interconnect experiments on 4- to 64-bit multipliers with both AND and Booth PPG. For each bit-width, 50 random interconnected designs are evaluated using the proxy delay model in UFO-MAC [10] as well as the actual post-synthesis delay. As shown in Figure 3, the discrepancy between proxy and actual delay metric increases with bit-width, underscoring the growing inaccuracy of proxy-driven optimization in large-scale designs.

To address these challenges and fully exploit the vast yet underexplored design space of compressor interconnections, we propose ARITH-DAS, a proxy-free differentiable architecture search framework based on graph neural networks. ARITH-DAS optimizes complex compressor-to-compressor routing by enabling accurate and efficient interconnect prediction aligned with post-synthesis metrics. It can be seamlessly integrated with any allocation optimization algorithm, forming a unified and extensible framework for circuit-level design optimization.

## 4 Methodology

We begin with an overview of our proposed ARITH-DAS in Section 4.1. Next, we propose our architecture search formulation in Section 4.2, including detailed multi-relational graph representation in Section 4.2.1 and differentiable relaxation in Section 4.2.2. Finally, Section 4.3 provides a detailed description of the core components of our framework, along with the training procedure aligned with post-synthesis performance metrics.

### 4.1 Overview of Our Framework

As illustrated in Figure 4, our ARITH-DAS framework consists of three principal components. To thoroughly explore the design space of arithmetic circuits, we first construct a coarse-grained compressor allocation using an adaptable circuit evolutionary strategy. This initial configuration is then expanded into a multi-relational directed acyclic graph (DAG), which defines a rich and expressive interconnect search space. A multi-relational graph neural network is employed to encode both topological and semantic relationships among architectural components, while a graph attention mechanism estimates the probabilistic significance of candidate connections. Finally, the entire framework is trained end-to-end via a PPO-inspired algorithm, enabling proxy-free optimization directly guided by post-synthesis performance metrics.

### 4.2 Architecture Search Formulation for Arithmetic Circuit

#### 4.2.1 Graph Representation for Compressor Tree

Combinational logic circuits can be naturally represented as directed acyclic graphs (DAGs), exhibiting structural equivalence between the circuit and its graph-based abstraction. In our representation, partial products and compressors are modeled as nodes, and logical dependencies are modeled as directed edges, as illustrated in Figure 1(c). To address the inherent complexity of fine-grained interconnect modeling, we employ a principled graph construction strategy, detailed below.

**(1) Topology Ordering via Stage Assignment** To preserve acyclicity, a fundamental constraint of combinational logic, we assign each compressor to a specific stage and allow interconnections only between stages of increasing order. This stage-wise ordering induces a valid acyclic circuit topology and establishes explicit dataflow dependencies.

**(2) Virtual Nodes for Cross-Stage Connections** To address mismatches in input and output port counts that necessitate signal bypassing, we introduce virtual nodes, which serve as auxiliary 1:1 compressor-like entities that propagate signals across stages. This mechanism enforces stage-wise connectivity while preserving topological consistency and simplifying graph construction.

**(3) Multi-Relational Graph for Asymmetric Input Semantics** While addition is logically commutative, the compressor input ports are physically asymmetric. To capture this asymmetry, we represent the compressor tree as a multi-relational graph [16–18], where each edge type corresponds to a specific input port of the target node. This formulation enables the model to distinguish structurally similar yet semantically distinct connections.

More formally, we define the compressor tree as a multi-relational directed acyclic graph [19], denoted by the tuple $(\mathcal{V}, \mathcal{E}, \mathcal{R})$. Here, $\mathcal{V} = \{v_1, v_2, \dots\}$ is the set of nodes, each representing a circuit

element. $\mathcal{R} = \{r_1, r_2, \dots\}$ is the set of relation types, where each $r \in \mathcal{R}$ corresponds to a semantically distinct input port. The edge set $\mathcal{E} \subseteq \mathcal{V} \times \mathcal{R} \times \mathcal{V}$ consists of typed directed edges, where each edge $(v_i, r, v_j) \in \mathcal{E}$ indicates that node $v_i$ connects to input port $r$ of node $v_j$. The graph can also be encoded as a three-dimensional binary tensor $\mathcal{G} \in \{0,1\}^{|\mathcal{R}| \times |\mathcal{V}| \times |\mathcal{V}|}$, where $\mathcal{G}_{r,i,j} = 1$ if and only if $(v_i, r, v_j) \in \mathcal{E}$. Following previous works [3, 7, 15, 20, 21], we represent the allocation of a compressor tree by a matrix $\mathbf{s} \in \mathbb{N}+^{T \times N}$, where $T$ is the number of compressor types and $N$ is the input bit width of the compressor tree. Each entry $\mathbf{s}_{t,n}$ denotes the number of compressors of type $t$ assigned to column $n$. We denote the set of valid designs by $\mathbb{G}$ and the set of valid structures by $\mathbb{S}$. The detailed design constraints are provided in Appendix C.3.2. Finally, our goal is to maximize a composite objective function:

$$\max_{\mathcal{G} \in \mathbb{G}} R(\mathcal{G}) = -w_1 \cdot \mathrm{area}(\mathcal{G}) - w_2 \cdot \mathrm{delay}(\mathcal{G}), \qquad (1)$$

where the objective function $R : \mathbb{G} \to \mathbb{R}$ is the weighted linear combination of post-synthesis area and delay metrics following prior works [3, 7, 9, 15].

### 4.2.2 Differentiable Reformulation for Discrete Search Space

Problem (1) constitutes a large-scale combinatorial optimization challenge, involving highly irregular graph structures and complex design constraints. To address this, we draw inspiration from differentiable architecture search [22–26], which offers a scalable and efficient alternative. By relaxing discrete structural decisions into continuous probability distributions, these methods enable joint optimization over a vast design space within a single forward pass, thereby significantly enhancing search efficiency and scalability.

Specifically, we replace the hard edge assignments in the compressor tree graph $\mathcal{G} \in \mathbb{G}$ with soft, learnable probability distributions over candidate source nodes. Each potential connection is parameterized by a real-valued score and passed through a softmax function, yielding a continuous relaxation $\tilde{\mathcal{G}} \in [0,1]^{|\mathcal{R}| \times |\mathcal{V}| \times |\mathcal{V}|}$, where $\tilde{\mathcal{G}}_{r,i,j}$ denotes the probability of establishing an edge from node $v_i$, conditioned on edge type $r$ and target node $v_j$, with normalization $\sum_i \tilde{\mathcal{G}}_{r,i,j} = 1$ for each $(r, j)$ pair. This target-centric formulation arises from the structural constraints and multi-relational graph design, which are inherently defined from the perspective of the receiving node. By modeling connection probabilities in this manner, each input port selects its optimal driver from valid candidates, enabling fine-grained, port-specific connectivity through softmax-based selection. This relaxation implicitly defines a distribution $\pi : \mathbb{G} \to [0,1]$ over the graph search space, with corresponding parameter space $\Pi$. Following previous works[22, 23], the objective is relaxed to

$$\max_{\pi \in \Pi} \mathbb{E}_{\mathcal{G} \sim \pi}\big[R(\mathcal{G})\big] = \max_{\mathbf{s} \in \mathbb{S}} \max_{\pi(\cdot|\mathbf{s}) \in \Pi_{\mathbf{s}}} \mathbb{E}_{\mathcal{G} \sim \pi(\cdot|\mathbf{s})}\big[R(\mathcal{G})\big], \qquad (2)$$

where $\Pi_{\mathbf{s}} \subset \Pi$ denotes the set of graph distributions associated with a given allocation $\mathbf{s}$. This naturally forms a two-stage optimization framework where we first determine the allocation of the compressor tree, and next we optimize the corresponding interconnection.

### 4.3 ARITH-DAS : Differentiable Architecture Search for Arithmetic Circuit

In this section, we present ARITH-DAS , a novel differentiable architecture search framework tailored to arithmetic circuit optimization. By encoding structural information via a multi-relational graph neural network and modeling interconnect prediction through graph attention, it enables end-to-end, gradient-based optimization directly guided by post-synthesis performance metrics.

### 4.3.1 Adaptable Allocation Search via Circuit Evolution

The compressor allocation $\mathbf{s} \in \mathbb{S}$ specifies the number of compressors across columns, shaping the coarse-grained structure and reduction stages of the compression tree. This aspect has been extensively studied [3, 7, 9, 10, 15, 20], with methods ranging from reinforcement learning to mixed-integer programming for optimizing compressor configurations under fixed encoding schemes.

To ensure simplicity, extensibility, and structural diversity, we adopt the evolution-based method introduced in [7]. Specifically, we maintain a population of candidate architectures in an elite pool, each representing a compressor allocation matrix across all columns. New candidates are generated through two key mechanisms: (1) *local perturbation*, where a small number of compressors are randomly added, removed, or shifted across columns, and (2) *substructure crossover*, where two parent architectures exchange subregions of their compressor allocation matrices.

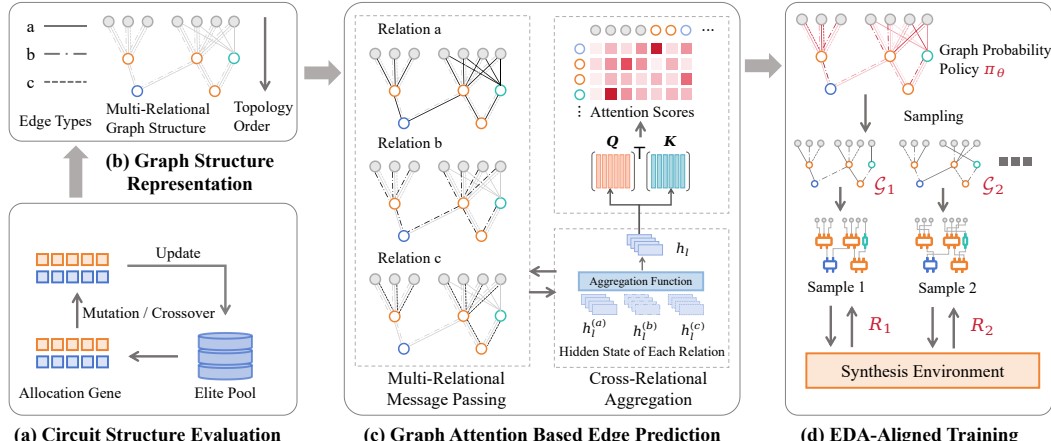

Figure 4: Overview of our proposed ARITH-DAS framework.

This evolutionary process promotes exploration of diverse architectural patterns while preserving high-performing solutions across generations. Crucially, it operates independently of the interconnect structure, enabling seamless integration with the differentiable wiring search described in the next stage. Once a compressor allocation configuration is sampled, we construct the corresponding multi-relational graph and optimize its interconnects accordingly.

### 4.3.2 Multi-Relational Graph Attention Based Link Prediction

**Multi-Relational Graph Encoder** Given a compressor allocation $\mathbf{s} \in \mathbb{S}$, our objective is to determine its optimal interconnect configuration. To this end, we leverage a multi-relational graph neural network to encode the underlying circuit topology and model the propagation of physical information through the circuit. Each edge type captures a distinct semantic relation, such as a specific input port of a compressor, and is processed independently during message passing, enabling the network to learn asymmetric and relation-specific connection patterns. Concretely, we adopt a multi-relational message passing scheme. At layer $l$, the hidden state of node $v_i$ under relation $r$ is updated by:

$$h_{i,r}^{l+1} = \sum_{j \in \mathcal{N}_r(i) \cup \{i\}} \frac{1}{\sqrt{|\mathcal{N}_r(i)|} \cdot \sqrt{|\mathcal{N}_r(j)|}} \phi_r^l \left(h_i^l, h_j^l\right), \ r \in \mathcal{R}, \tag{3}$$

where $\mathcal{N}_r(i) = \{v_j \in \mathcal{V} \mid (v_i, r, v_j) \in \mathcal{E} \text{ or } (v_j, r, v_i) \in \mathcal{E}\}$ denotes the set of nodes bidirectionally connected to $v_i$ under relation $r$, and $\phi_r^l$ is a relation-specific message function at layer $l$, augmented with self-loops and reverse edges to support comprehensive message flow, while directional semantics are embedded in node features. To integrate information across all relations, we concatenate the relation-specific representations and apply a shared aggregation function $\gamma^l$:

$$h_i^{l+1} = \gamma^l \left(\text{Concat}_{r \in \mathcal{R}}(h_{i,r}^{l+1})\right). \tag{4}$$

**Graph Attention Based Edge Prediction** The final node embedding $h_i$ encodes the structural context of vertex $v_i$. To predict interconnect probabilities, we employ a Graphormer-style attention mechanism [27], wherein each target node and its candidate source nodes are projected into relation-specific query and key spaces, respectively. Let $\mathcal{O} = \{\text{sum}, \text{carry}\}$ denote the set of output port types. Given a target node $v_j$ and a port pair $(o, r) \in \mathcal{O} \times \mathcal{R}$, the attention score from a candidate source node $v_i$ is computed as:

$$\alpha_{i,j}^{o,r} = \frac{\exp\left\{(W_Q^r h_j)^\top (W_K^o h_i)\right\} \cdot \mathcal{M}_{i,j}^{o,r}}{\sum_{i' \in [|\mathcal{V}|]} \exp\left\{(W_Q^r h_j)^\top (W_K^o h_{i'})\right\} \cdot \mathcal{M}_{i',j}^{o,r}}, \tag{5}$$

where $W_Q^r$ and $W_K^o$ are linear projections specific to the input and output port types, respectively. The binary mask $\mathcal{M}_{i,j}^{o,r} \in \{0, 1\}$ imposes structural constraints by limiting the softmax normalization to valid source-target-port combinations, as detailed in Appendix C.3.2. This formulation conforms to the probabilistic semantics of our relaxation: for each target node and input port type, it defines a distribution over valid source nodes. The target node acts as the query and the source node as the key, ensuring consistency between the attention logits and relaxed edge selection probabilities.

### 4.3.3 Post-Synthesis Alignment via Proxy-Free PPO-Like Training

Given a fixed compressor allocation $\mathbf{s}$, the attention scores $\alpha_{i,j}^{o,r}$ define a structured distribution over interconnect candidates. A valid discrete realization is obtained via sequential sampling along a legal topological order, with the legality mask $\mathcal{M}_{i,j}^{o,r}$ dynamically updated to ensure acyclicity, stage consistency, and port compatibility. This process yields a deterministic graph sample $\mathcal{G} \sim \pi_\theta(\cdot \mid \mathbf{s})$, where the probability is factorized as $\pi_\theta(\mathcal{G} \mid \mathbf{s}) = \prod_{r,i,j,o} \mathcal{G}_{r,i,j} \cdot \alpha_{i,j}^{o,r}$, fully parameterized by the learnable attention logits $\alpha_{i,j}^{o,r}$. To optimize the interconnect structure with respect to post-synthesis performance metrics, we employ a proxy-free, PPO-like training paradigm [28]. Specifically, we treat $\pi_\theta$ as a stochastic policy over the graph space and maximize the expected objective defined by post-synthesis metrics $R$:

$$\max_\theta J(\theta \mid \mathbf{s}) = \max_\theta \mathbb{E}_{\mathcal{G} \sim \pi_\theta(\cdot \mid \mathbf{s})} \big[ R(\mathcal{G}) \big] = \max_\theta \mathbb{E}_{\mathcal{G} \sim \pi_{\overline{\theta}}(\cdot \mid \mathbf{s})} \left[ \frac{\pi_\theta(\mathcal{G} \mid \mathbf{s})}{\pi_{\overline{\theta}}(\mathcal{G} \mid \mathbf{s})} R(\mathcal{G}) \right], \tag{6}$$

where $\pi_{\overline{\theta}}$ is a fixed reference policy used for importance sampling. To enforce conservative updates and prevent large deviations from the reference policy, we optimize the clipped surrogate objective:

$$\hat{J}(\theta \mid \mathbf{s}) = \frac{1}{M} \sum_{m=1}^M \min \left( \rho_m R(\mathcal{G}_m), \ \mathrm{clip}(\rho_m, 1 - \epsilon, 1 + \epsilon) R(\mathcal{G}_m) \right), \tag{7}$$

where $\rho_m = \frac{\pi_\theta(\mathcal{G}_m \mid \mathbf{s})}{\pi_{\overline{\theta}}(\mathcal{G}_m \mid \mathbf{s})}$ is the likelihood ratio between the current and reference policies, and $\mathcal{G}_m \sim \pi_{\overline{\theta}}(\cdot \mid \mathbf{s})$ are i.i.d. sampled graphs. The hyperparameter $\epsilon$ controls the trust region for updates. Additionally, we introduce a regularization term to promote structural discreteness and enforce port-level sparsity constraints of the output ports [11]. The overall loss is defined as:

$$L(\theta \mid \mathbf{s}) = \underbrace{\sum_{i,j \in [|\mathcal{V}|], o \in \mathcal{O}, r \in \mathcal{R}} \left[ \alpha_{i,j}^{o,r} \cdot (1 - \alpha_{i,j}^{o,r}) \right]^2}_{\text{Discretization penalty}} + \underbrace{\sum_{i \in [|\mathcal{V}|], o \in \mathcal{O}} \left( 1 - \sum_{j \in [|\mathcal{V}|], r \in \mathcal{R}} \mathcal{M}_{r,i,j} \cdot \alpha_{i,j}^{o,r} \right)^2}_{\text{Output-port exclusivity constraint}} .$$
$$\tag{8}$$

The first term penalizes probabilities that lie in the ambiguous region $(0, 1)$, thus promoting discrete-like selection. The second term enforces that each output port selects exactly one target input (i.e., normalized to 1). Finally, the complete training objective combines the PPO-style reward maximization with the regularization:

$$\max_\theta \mathcal{L}(\theta) = \mathbb{E}_{\mathbf{s} \sim \mathbb{S}} \left[ \hat{J}(\theta \mid \mathbf{s}) - \eta L(\theta \mid \mathbf{s}) \right], \tag{9}$$

where $\eta > 0$ controls the trade-off between reward fidelity and structural regularity. A theorem-guided discussion is detailed in Appendix C.3.6.

## 5 Experiments

We begin by describing the experimental setup, baseline methods, and evaluation metrics in Section 5.1. The experiments are designed to pursue three primary objectives: **(1)** Evaluate the effectiveness of ARITH-DAS in optimizing computing multipliers and MAC units across a broad range of input bit-widths (Section 5.2); **(2)** Assess the generalization ability of ARITH-DAS -optimized multipliers when scaled to large macro designs representative of real-world AI accelerators (Section 5.3); **(3)** Perform ablation studies to quantify the contributions of individual components within ARITH-DAS and elucidate the rationale behind its design (Section 5.4).

### 5.1 Experiment Setup

**Experimental Setup** Our framework leverages OpenROAD [29] for physical implementation. Logic synthesis is performed using Yosys [30] with the Nangate45 technology library [31], while static timing analysis (STA) is conducted through OpenSTA [29]. Functional verification employs Verilator [32] for cycle-accurate simulation. The machine learning pipeline is implemented in PyTorch [33] and PyTorch Geometric [34], optimized via the Adam algorithm [35]. We evaluate our methodology across eight distinct multiplier architectures, encompassing 8-bit, 16-bit, 32-bit, and 64-bit implementations employing both AND gate-based and Booth encoding-based techniques. More experiment configurations are provided in Appendix C.

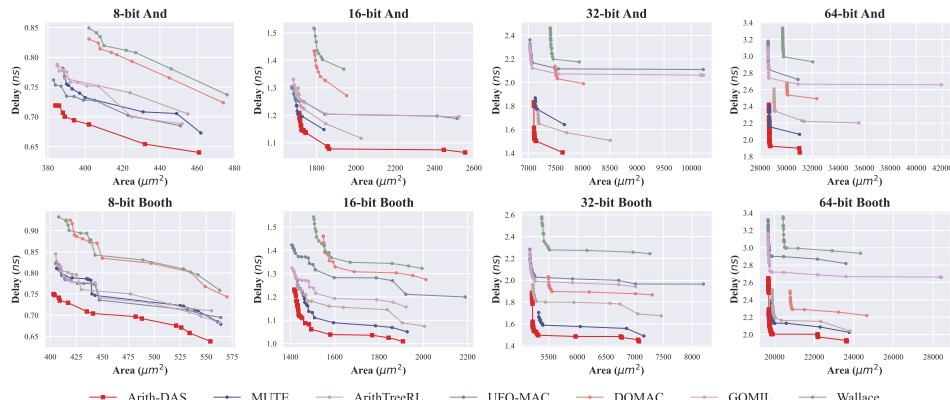

Figure 6: Pareto frontiers of our ARITH-DAS and baselines on eight multiplier design tasks.

**Comparative Baselines** We evaluate our methodology against six representative approaches spanning classical heuristic designs to contemporary learning-to-optimize (L2O) paradigms: **(1) Wallace** [36]: A foundational heuristic compression technique; **(2) UFO-MAC** [10] and **(3) GOMIL** [20]: A solution framework based on integer programming; **(4) DOMAC** [9]: A differentiable interconnection optimization framework; **(5) ArithTreeRL** [9]: A leading deep reinforcement learning (RL) architecture; **(6) MUTE** [7]: The current state-of-the-art hybrid RL-Evolutionary framework. Extended comparative analyses including implementation specifics and hyperparameter configurations are provided in Appendix C.4.

**Evaluation Metrics** Arithmetic circuit optimization constitutes a canonical multi-objective optimization problem. We employ two established evaluation protocols from multi-objective optimization theory: **(1) Pareto Frontier Analysis:** Following established methodologies [3, 7–11, 15], we simulate diverse design preferences through parametric target delay configurations. This enables comprehensive synthesis of corresponding solutions and visualization of their Pareto frontiers [37]. **(2) Hypervolume (HV) Metric:** We conduct comparative analysis of solution quality through hypervolume measurements [38] across different Pareto frontiers, quantifying their multi-objective characteristics. More details are provided in Appendix C.3.8.

## 5.2 Main Evaluation

We highlight the superiority of ARITH-DAS through a comparative analysis with six competitive baselines on eight multiplier design problems across a wide range of input sizes. The results in Figure 6 demonstrate that multipliers optimized by ARITH-DAS consistently and significantly outperform designs optimized by all baselines across all eight multiplier design tasks. Moreover, we present the hypervolume (HV) of the Pareto points discovered by ARITH-DAS in Table 1. The results demonstrate that ARITH-DAS achieves a substantial improvement over the previous SOTA, improving the hypervolume by up to 27.05%. Overall, these results demonstrate the strong ability of ARITH-DAS to optimize multipliers, leading to significant reductions in both area and delay. More results, including evaluation on multiply-accumulators (MACs) and visualization of results, are provided in Appendix D.

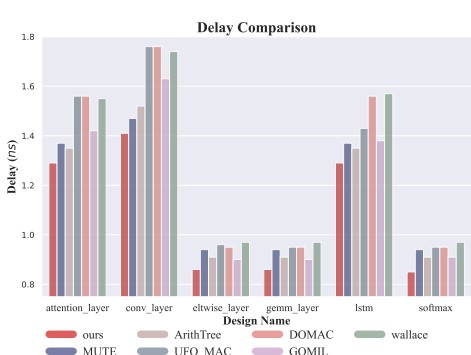

Figure 5: Delay of practical AI chips with our ARITH-DAS and baselines integrated.

## 5.3 Generalization to Large-Scale Computing Circuit

To assess the generalization capability of the optimized computing units, we integrate those produced by ARITH-DAS and baseline methods into six representative AI accelerator circuits from the Koios 2.0 benchmark [39]. In addition, we deploy and evaluate these units within a typical processing element (PE) array architecture, which is widely used in applications, following previous works [7, 15, 21]. As shown in Figure 5 and Figure 7, circuits incorporating ARITH-DAS -optimized units consistently outperform these baseline counterparts across multiple metrics. These

Table 1: Hypervolume (HV.) of multiplier design.

| Mult. (AND) | 8 bit | | 16 bit | | 32 bit | | 64 bit | |
|---|---|---|---|---|---|---|---|---|
| Method | HV. ↑ | Impr.(%) | HV. ↑ | Impr.(%) | HV. ↑ | Impr.(%) | HV. ↑ | Impr.(%) |
| Wallace | 3.43 | NA | 107.53 | NA | 793.19 | NA | 4,796.14 | NA |
| GOMIL | 11.56 | 237.25 | 265.17 | 146.60 | 1,216.48 | 53.36 | 8,784.55 | 83.16 |
| UFO-MAC | 12.83 | 274.08 | 269.18 | 150.32 | 1,076.70 | 35.74 | 7,932.19 | 65.39 |
| DOMAC | 4.60 | 34.29 | 177.52 | 65.09 | 1,267.22 | 59.76 | 9,941.63 | 107.28 |
| ArithTreeRL | 9.61 | 180.29 | 315.19 | 193.11 | 2,779.56 | 250.43 | 14,237.63 | 196.86 |
| MUTE | 11.67 | 240.30 | 310.89 | 189.11 | 2,470.47 | 211.46 | 16,656.60 | 247.29 |
| ARITH-DAS (Ours) | 16.30 | **375.32** | 361.84 | **236.49** | 3,247.06 | **309.37** | 19,563.48 | **307.90** |
| Mult. (Booth) | 8 bit | | 16 bit | | 32 bit | | 64 bit | |
| Method | HV. ↑ | Impr.(%) | HV. ↑ | Impr.(%) | HV. ↑ | Impr.(%) | HV. ↑ | Impr.(%) |
| Wallace | 14.92 | NA | 130.88 | NA | 878.23 | NA | 3,146.58 | NA |
| GOMIL | 31.50 | 111.16 | 274.09 | 109.43 | 1,852.58 | 110.95 | 5,842.81 | 85.69 |
| UFO-MAC | 31.50 | 111.18 | 211.36 | 61.49 | 1,754.11 | 99.73 | 4,408.90 | 40.12 |
| DOMAC | 15.54 | 4.17 | 152.25 | 16.33 | 1,875.59 | 113.57 | 8,440.55 | 168.25 |
| ArithTreeRL | 29.94 | 100.72 | 311.13 | 137.88 | 2,455.28 | 179.57 | 10,728.16 | 240.95 |
| MUTE | 30.61 | 105.15 | 346.68 | 164.89 | 2,961.62 | 237.23 | 11,034.68 | 250.69 |
| ARITH-DAS (Ours) | 39.94 | **167.70** | 389.63 | **197.71** | 3,305.51 | **276.38** | 12,212.89 | **288.13** |

Figure 7: Pareto front of PE arrays integrated with designs generated by ARITH-DAS and baselines.

results demonstrate the strong generalization ability of ARITH-DAS in large-scale, computation-intensive circuits, underscoring its potential to enhance the performance of real-world AI chips.

## 5.4 Ablation Study

We conducted a carefully designed ablation study targeting the multiplier design task. As has been noted, our method is composed of the following key modules: **(1)** the Circuit Genetic Evolution (CGE) module, **(2)** the Multi-Relational Graph Encoder (MRG) module, and **(3)** the PPO-style training module (PPO). To assess the individual contribution of each module within ARITH-DAS , we conduct a

Table 2: Ablation study

| Mult. (AND) | 16 bit | | 32 bit | |
|---|---|---|---|---|
| Method | HV. ↑ | Impr.(%) | HV. ↑ | Impr.(%) |
| Wallace | 107.53 | NA | 793.19 | NA |
| ARITH-DAS (Ours) | 361.84 | 236.49 | 3,247.06 | 309.37 |
| w/o CGE | 326.58 | 203.71 | 2427.64 | 206.06 |
| w/o MRG | 313.57 | 191.61 | 2499.68 | 215.14 |
| w/o PPO | 326.49 | 203.63 | 3007.14 | 279.12 |

comprehensive component-wise analysis focused on the optimization of multiplier architectures. Specifically, we design three ablation experiments by removing each module, respectively, to quantify its impact on overall performance: **(1) w/o CGE** where circuit genetic evolution is replaced with a simulated annealing approach; **(2) w/o MRG** where the multi-relational graph encoder is replaced by a heterogeneous-node graph with uniform edge types; **(3) w/o PPO** where the PPO-style loss is replaced by the proxy delay model introduced in previous work [11]. As shown in Table 2, all three modules are critical to the overall performance. Removing the CGE module impairs structural exploration and reduces architectural diversity, underscoring the role of genetic search in design space navigation. Removing the MRG module weakens relational reasoning, confirming the necessity of modeling edge semantics for accurate connectivity inference. Substituting the PPO-based optimization with a proxy delay model degrades performance, highlighting the benefit of end-to-end training aligned with post-synthesis metrics.

## 6 Conclusion and Limitations

In this work, we propose ARITH-DAS , a differentiable architecture search framework for arithmetic circuit optimization. Experiments on representative arithmetic units show that ARITH-DAS consistently outperforms state-of-the-art baselines in area-delay trade-offs. When deployed in large-

scale AI accelerators, it achieves notable timing improvements, demonstrating strong scalability and practical value. These results underscore the effectiveness of ARITH-DAS and offer new directions for optimizing high-performance computing systems.

However, there still remain several limitations. Consistent with prior studies, our current experimental setup remains confined to the post-synthesis stage, which does not account for the intricate backend procedures such as placement, routing, and timing closure. These stages have a profound impact on the circuits final area, timing, and power characteristics, often leading to substantial discrepancies between synthesized and implemented results. Hence, exploring backend-aware optimization strategies for arithmetic circuits constitutes an essential avenue for future research.

## 7 Acknowledgements

We would like to thank all the anonymous reviewers for their insightful comments. The numerical calculations in this paper have been done on the supercomputing system in the Supercomputing Center of the University of Science and Technology of China. This work was supported by the National Key R&D Program of China under contract 2022ZD0119801, and the National Natural Science Foundation of China grants U23A20388, 62021001, and 624B1011. We would like to thank our lab-mate, Zhaojie Tu, for his valuable technical support.

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

# A  Related Work

## A.1  Computer arithmetic

Computational circuits such as adders and multipliers serve as foundational components in modern computing systems, enabling high-performance parallel operations through advanced architectures such as compression trees and prefix trees. The optimization methodologies for these circuits predominantly fall into three categories: (1) **Manual Design**: This approach relies on domain-specific knowledge to refine conventional architectures, demanding considerable engineering resources. Notable implementations include partial product reduction techniques using compression tree [36, 40–42], and optimized parallel addition through prefix tree [43–45]. (2) **Traditional Algorithmic Approaches** ([10, 20, 46, 47]): These methods employ mathematical programming and heuristic search to generate circuit architectures. However, their dependence on proxy metrics (e.g., size, depth) frequently introduces substantial deviations from physical implementation outcomes. (3) **Learning-Based Approaches** [3, 8, 15, 21, 48, 49]: Emerging techniques integrate synthesis processes with reinforcement learning frameworks, directly targeting post-synthesis metrics to mitigate the fidelity loss inherent in proxy-based optimization. RL-MUL [21] first proposed a reinforcement learning framework for multiplier optimization, and HAVE [15] later extended it into a hierarchical multi-task RL framework. Recent work, DOMAC [11], introduced a differentiable interconnection optimization algorithm based on an estimated proxy delay model, offering a new perspective for arithmetic circuit optimization. In this work, we similarly focus on optimizing interconnection within arithmetic circuits using post-synthesis metrics.

## A.2  Machine learning for chip design

As semiconductor processes advance into the nanoscale era, integrated circuit complexity grows exponentially, creating computational complexity bottlenecks for traditional EDA tools in PPA optimization. Machine learning-based design automation methodologies, particularly Graph Neural Networks (GNNs), have emerged as a novel paradigm to address these challenges. The canonical chip design workflow encompasses critical stages including logic synthesis, floorplanning, clock tree synthesis, and routing. The netlist, serving as an intermediate representation compiled from hardware description languages (HDLs), inherently embodies a graph-based data structure that encapsulates gate-level circuit components and their directed interconnect dependencies. This intrinsic graph topology makes GNNs particularly effective for extracting topological features, as evidenced by several pioneering studies, such as LaMPlace[50], TP-GNN[51], and TA3D[52]. There is a publicly available benchmark for AI solutions in backend layout and routing [53].

## A.3  Differentiable architecture search

Recently, differentiable architecture search has been getting a lot of attention in a variety of areas. In the domain of neural architecture search (NAS), DARTS[22] pioneered a gradient-based approach by relaxing discrete architectures into continuous spaces, enabling efficient optimization through gradient descent. Building upon this, ProxylessNAS [23] directly searches for neural network architectures on the target tasks and hardware, addressing the high computational cost of traditional NAS and the large memory consumption of differentiable NAS. In the domain of logic gate network architecture search, Petersen et al. [24] resolves the non-differentiability issue of logic gate networks via real-valued logic and continuous relaxation, enabling them to be trained using gradient descent. Further extending this direction, Petersen et al. [54] presents Convolutional Differentiable Logic Gate Networks, which enhance Differentiable Logic Gate Networks through deep logic gate tree convolutions, logical OR pooling, and residual initialization. In the domain of hardware optimization, Wang et al. [25] proposed a neural circuit generation framework using differentiable architecture search and reinforcement learning, enabling precise synthesis of circuits with large-scale nodes while outperforming traditional logic synthesis tools.

## A.4  Graph Neural Networks

GNNs learn on graphs via iterative neighborhood aggregation [55, 56], and are applied broadly in graph-related problems [57–60]. Canonical architectures, GCN (spectral/spatial convolution) [55], GraphSAGE (inductive sampling) [61], GAT (attention over neighbors) [62], and GIN (WL-

strengthened expressivity) [63], define the fields expressivity-scalability trade-offs. Depth is limited by over-smoothing and over-squashing, motivating residual/skip connections, normalization, and multi-scale propagation [64, 65]. Beyond homogeneous graphs, multi-relational/heterogeneous GNNs parameterize typed edges and meta-relations: R-GCN [19] and CompGCN [16] perform relation-specific propagation, while HAN/HGT incorporate relation-aware attention for web-scale heterogeneity [66, 67]. In parallel, graph Transformers (e.g., Graphormer) inject structural biases into global attention to capture long-range dependencies [27]. There is also a recent survey of reinforcement learning on graphs [68].

## B  Design Space Complexity Analysis

In this section, we analyze the complexity of the design space of the compressor tree allocation and the compressor interconnection. Previous works [3, 9] have shown the $O(2^{N^2})$ complexity for compressor tree allocation design, where $N$ is the input bitwidth of the arithmetic circuit. Here, we provide a complexity analysis of the interconnection space.

Without loss of generality, let's take the AND PPG multiplier as an example. First, the number of partial products of the AND PPG multiplier with an input bit-width of $N$ is exactly $N^2$. Next, a 3:2 compressor compresses one partial product while a 2:2 compressor doesn't compress. Therefore, at least $O(N^2)$ 3:2 compressors are needed to compress the $O(N^2)$ partial products into compressed products with complexity of $O(N)$. Finally, the complexity of the shape of a 3-relational directed graph representation of the compressor tree is at least $\sim 3 \times N^2$, which indicates an $O(2^{3N^2})$ search space. Note that, since the stage assignment determines the topological order of nodes and constrains compression to occur within each stage, interconnection happens only locally. As a result, connections are established only locally, and the complexity of the interconnection space is linear rather than quadratic with compressor number.

Overall, the design space complexity of a compressor tree, considering both its allocation and interconnection, exhibits an extraordinary growth rate of $O(2^{4N^2})$.

## C  Implementation Details

### C.1  Hardware Specification

All experiments are executed on a computational platform with an Intel Xeon Gold 6246R CPU (3.60 GHz) and NVIDIA RTX 3090 GPU.

### C.2  Synthesis Tool Setup

We use Yosys [30] for logic synthesis and OpenSTA [69] for STA simulation in conjunction with Nangate45 open cell library [31], which is widely used in the semiconductor industry and related research. Synthesis script and SDC constraints are provided at Listing 1 and Listing 2.

All RTL designs generated by our method and baselines are functionally verified with Verilator [32]. We randomly generate 10000 input samples, and perform a bit-wise comparison between the Verilator evaluated result and the ground truth calculated by C++ codes.

Listing 1: SDC Constraints

```
1  set_driving_cell BUF_X1
2  set_load 10.0 [all_outputs]
```

```
1  read -sv {verilog_path}
2  synth -top {top_name}
3  dfflibmap -liberty {liberty_path}
4  abc -D {target_delay} -constr {constr_path} -liberty {liberty_path}
5  write_verilog {netlist_path}
```

### C.3 Implementation Details of Our ARITH-DAS

#### C.3.1 Further Discussion on Multi-Relational Graph Formulation

In the main paper, we formulate the connection as a multi-relational directed acyclic graph $(\mathcal{V}, \mathcal{E}, \mathcal{R})$ and corresponding adjacent tensor $\mathcal{G} \in \{0, 1\}^{|\mathcal{R}| \times |\mathcal{V}| \times |\mathcal{V}|}$. More implementation-specific details are provided in this section.

**Vertex Set** Vertex set is defined as $\mathcal{V} = \{v_1, v_2, \dots\}$, where a vertex $v_i$ corresponds to one of the following four basic components: **(1)** 3:2 compressor, which is a full adder, **(2)** 2:2 compressor, which is a half adder, **(3)** initial partial product, which is the output of PPG and **(4)** visual node, which is a wire. For the sake of implementation convenience, we also define output nodes as virtual nodes within our graph representation. This unification allows for a consistent treatment of all nodes during message passing and connectivity inference. Each node belongs to a specific column and is assigned to a specific stage during stage assignment [3]. The column number and the stage number of nodes topologically specify their interconnection order, i.e., from lower column to higher column, and from lower stage to higher stage. This approach leverages the homogeneity among nodes to significantly reduce technical complexity.

**Relational Set** Relational set is defined as $\mathcal{R} = \{r_1, r_2, \dots\}$, where each relation type $r_i$ represents an input port type. In this work we use 3:2 compressors and 2:2 compressors, and therefore $\mathcal{R} = \{a, b, cin\}$. For 2:2 compressors, only input ports $\{a, cin\}$ are available. For visual nodes, only $\{a\}$ is available. For PP nodes, no relation is available since they do not have input ports. The relational types are specified according to the input port types, rather than the output port types, mainly due to the fact that the input ports are logically indistinguishable, while the output ports are connected to different columns. Accordingly, we aim for each node to learn the semantics of its output connections through training.

**Edge Set and adjacent tensor** Typed directed edge set is defined as $\mathcal{E} \subset \mathcal{V} \times \mathcal{R} \times \mathcal{V}$, where each edge $(v_i, r, v_j)$ represents a directed logical wire from output port of $v_i$ connected to the $r$-th type input port of $v_j$. Since edges are type-specific, we construct a separate graph for each edge type, referred to as a graph channel. Specifically, the value of $\mathcal{G} \in \{0, 1\}^{|\mathcal{R}| \times |\mathcal{V}| \times |\mathcal{V}|}$ is defined as

$$\mathcal{G}_{r,i,j} = \begin{cases} 0 & (v_i, r, v_j) \notin \mathcal{E}, \\ 1 & (v_i, r, v_j) \in \mathcal{E}. \end{cases} \tag{10}$$

#### C.3.2 Architectural Design Constraints

In this section, we provide the detailed complex architectural constraints on the proposed multi-relational graph formulation. For node $v_i \in \mathcal{V}$, let $\mathrm{Col}(v_i)$ denote the column index of $v_i$, $\mathrm{Stage}(v_i)$ denote the stage index of $v_i$, and $\mathrm{Type}(v_i)$ denote the type of $v_i$. A legal design should satisfy the following constraints.

**Vertex types and relation types constraints** For any $(v_i, r, v_j) \in \mathcal{E}$, there should be $\mathrm{Type}(v_j) \neq$ pp and:

$$r \in \begin{cases} \{a, b, cin\} & \mathrm{Type}(v_j) = \text{3:2 compressor}, \\ \{a, cin\} & \mathrm{Type}(v_j) = \text{2:2 compressor}, \\ \{a\} & \mathrm{Type}(v_j) = \text{visual node}. \end{cases} \tag{11}$$

**Vertex typological constrains** For any $(v_i, r, v_j) \in \mathcal{E}$, there should be $\text{Stage}(v_j) = \text{Stage}(v_i) + 1$, $\text{Col}(v_j) = \text{Col}(v_i)$ for sum wires of $v_i$ and $\text{Col}(v_j) = \text{Col}(v_i) + 1$ for carry wires of $v_i$.

**Port exclusivity constrains** Denote the output node of the graph as the primary out (PO), and the input node of the graph as the primary in (PI). PO and PI nodes are specified according to the compressor allocation and stage assignment, which remain unchanged during the optimization process. Denote the set of primary in and primary out nodes as $PI$ and $PO$, respectively. In our settings, for any node $v \in PI$, $\text{Type}(v) = \text{pp}$; for any node $v \in PO$, $\text{Type}(v) = \text{visual node}$. Each output port of non-PO nodes should be connected only to one node, i.e., for any node $v_i$,

$$\sum_{r \in \mathcal{R}, j \in \text{Sum}[i]} \mathcal{G}_{r,i,j} = \begin{cases} 0 & v_i \in PO \\ 1 & v_i \notin PO \end{cases}, \qquad \sum_{r \in \mathcal{R}, j \in \text{Carry}[i]} \mathcal{G}_{r,i,j} = \begin{cases} 0 & v_i \in PO \\ 1 & v_i \notin PO \end{cases} \qquad (12)$$

where $\text{Sum}[i] = \{j \mid \text{Col}(v_j) = \text{Col}(v_i), \text{Stage}(v_j) = \text{Stage}(v_i) + 1\}$ is the set of nodes to which the sum out of node $v_i$ is legal, and $\text{Carry}[i] = \{j \mid \text{Col}(v_j) = \text{Col}(v_i) + 1, \text{Stage}(v_j) = \text{Stage}(v_i) + 1\}$ is the set of nodes to which the carry out of node $v_i$ is legal.

Similarly, the input port of a non-PI node should only receive a logical connection from one other node, i.e., for any node $v_j \in \mathcal{V}$ and relation $r \in \mathcal{R}$,

$$\sum_{i \in [|\mathcal{V}|]} \mathcal{G}_{r,i,j} = \begin{cases} 0 & v_j \in PI \text{ or } r \notin \text{Input}[j], \\ 1 & \text{else.} \end{cases} \qquad (13)$$

where $\text{Input}[j] \subseteq \mathcal{R}$ denotes the input types of node $v_j$.

**Legalization Mask** The legalization mask $\mathcal{M} \in \{0, 1\}^{|\mathcal{R}| \times |\mathcal{V}| \times |\mathcal{V}|}$ indicates whether a connection is legal. The value of $\mathcal{M}_{r,i,j}$ is:

$$\mathcal{M}_{r,i,j} = \begin{cases} 1 & \begin{matrix} r \in \text{Input}[j] \text{ and } \text{Stage}[v_j] = \text{Stage}[v_i] + 1 \\ \text{and} \left( \text{Col}[v_j] = \text{Col}[v_i] + 1 \text{ or } \text{Col}[v_j] = \text{Col}[v_i] \right) \end{matrix}, \\ 0 & \text{else.} \end{cases} \qquad (14)$$

### C.3.3 Multi-Relational Message Passing

To simulate the propagation of circuit information across different types of connections, we adopt a multi-relational message passing mechanism. Message passing occurs over all possible connection types, and each node aggregates incoming messages by merging hidden features from all relation-specific channels. It should be noted that the adjacency matrix used for message passing of relation $r \in \mathcal{R}$ is exactly the $r$-th channel of legalization mask $\mathcal{M}$. The aggregated representation is then passed through a shared single-layer MLP to produce the updated node state. It is worth noting that we do not explicitly fit any physical signal characteristics; instead, the entire process is trained in an end-to-end manner. Ablation studies confirm the necessity of this design choice.

### C.3.4 Graphformer-Style Attention Score Calculation

To improve runtime efficiency in such a large and sparse graph space, we compute attention scores and maintain validity masks in a localized manner. A caching mechanism is employed, whereby computations are only triggered upon cache misses, thus reducing unnecessary overhead.

### C.3.5 Regulized Sampling

Directly sampling from the raw probability distribution often leads to invalid structures. To address this, we adopt a regularized sampling strategy. Specifically, guided by the topological order provided by the stage assignment, we sample input ports for each node's output in sequence based on the connection probabilities. During sampling, a mask fill operation is applied to the attention scores prior to normalization to eliminate invalid connections. After each sampling step, the corresponding connection masks are dynamically updated to maintain structural legality, and probability logits are maintained for network updating. More specifically, when edge $(v_i, r, v_j) \in \mathcal{R}$ is sampled with $\mathcal{M}_{r,i,j} = 1$, we record corresponding $\alpha_{i,j}^{o,r}$ value and let $\mathcal{M}_{r,i',j'} = 0$ for all $i', j' \in [|\mathcal{V}|]$ and $\mathcal{M}_{r',i',j} = 0$ for all $i' \in [|\mathcal{V}|], r' \in \mathcal{R}$.

### C.3.6 Detailed Derivation and Discussion of the Loss Function

We first provide the detailed derivation of (6) as below. When given compressor tree allocation $\mathbf{s}$,

$$
\begin{aligned}
J(\theta \mid \mathbf{s}) &= \mathbb{E}_{\mathcal{G} \sim \pi_\theta(\cdot \mid \mathbf{s})}\big[R(\mathcal{G})\big] \\
&= \sum_{\mathcal{G} \in \mathbb{G}_\mathbf{s}} \pi_\theta(\mathcal{G} \mid \mathbf{s}) R(\mathcal{G}) \\
&= \sum_{\mathcal{G} \in \mathbb{G}_\mathbf{s}} \pi_{\overline{\theta}}(\mathcal{G} \mid \mathbf{s}) \frac{\pi_\theta(\mathcal{G} \mid \mathbf{s})}{\pi_{\overline{\theta}}(\mathcal{G} \mid \mathbf{s})} R(\mathcal{G}) \\
&= \mathbb{E}_{\mathcal{G} \sim \pi_{\overline{\theta}}(\cdot \mid \mathbf{s})} \left[ \frac{\pi_\theta(\mathcal{G} \mid \mathbf{s})}{\pi_{\overline{\theta}}(\mathcal{G} \mid \mathbf{s})} R(\mathcal{G}) \right].
\end{aligned}
\tag{15}
$$

Consequently, the unbiased statistical estimator of $J(\theta \mid \mathbf{s})$ is given by

$$
\tilde{J}(\theta \mid \mathbf{s}) = \frac{1}{M} \sum_{m=1}^M \frac{\pi_\theta(\mathcal{G}_m \mid \mathbf{s})}{\pi_{\overline{\theta}}(\mathcal{G}_m \mid \mathbf{s})} R(\mathcal{G}_m),
\tag{16}
$$

where $\mathcal{G}_m$ i.i.d. $\sim \pi_{\overline{\theta}}(\cdot \mid \mathbf{s})$. In PPO, the `clip` operation is typically introduced to ensure that the distributional shift between the sampling policy $\pi_{\overline{\theta}}$ and the updated policy $\pi_\theta$ does not become too large, which could otherwise lead to distorted surrogate objectives due to mismatched trajectory distributions [70].

However, in our setting, the problem can be reformulated as a single-step multi-armed bandit scenario rather than a sequential Markov Decision Process. Therefore, trajectory mismatch is not an issue. Nonetheless, the clipping mechanism remains crucial for reducing variance during policy updates. We formalize this intuition in the following proposition. First, we introduce Pearson $\chi^2$ divergence, which is widely used in machine learning [71, 72].

**Definition 1** (Pearson $\chi^2$ Divergence[73, 74]). *For two distribution $p$ and $q$, Pearson $\chi^2$ Divergence is defined as*

$$
\chi^2(p \| q) = \mathbb{E}_{x \sim q} \left[ \frac{p(x)}{q(x)} - 1 \right]^2.
\tag{17}
$$

Then, we have the following proposition.

**Proposition 1.** *Assume that the objective function $R$ is bounded:* $\max_{\mathcal{G} \in \mathbb{G}} |R(\mathcal{G})| \le R_m$. *Then the variance of estimator of $J(\theta \mid \mathbf{s})$ defined in (16) is bounded by $\chi^2$ divergence:*

$$
\mathrm{Var}_{\pi_{\overline{\theta}}}(\tilde{J}(\theta \mid \mathbf{s})) \le \frac{1}{M} \left[ R_m^2 \Big( 1 + \chi^2 \big( \pi_\theta(\cdot \mid \mathbf{s}) \big\| \pi_{\overline{\theta}}(\cdot \mid \mathbf{s}) \big) \Big) - J^2(\theta) \right].
\tag{18}
$$

*Proof.* Let $\rho_\theta(\mathcal{G} \mid \mathbf{s}) = \frac{\pi_\theta(\mathcal{G}_m \mid \mathbf{s})}{\pi_{\overline{\theta}}(\mathcal{G}_m \mid \mathbf{s})}$. Since samples $\mathcal{G}_m$ are i.i.d., then

$$
\begin{aligned}
\mathrm{Var}_{\pi_{\overline{\theta}}(\cdot \mid \mathbf{s})}(\tilde{J}(\theta \mid \mathbf{s})) &= \mathrm{Var}_{\pi_{\overline{\theta}}(\cdot \mid \mathbf{s})} \left[ \frac{1}{M} \sum_{m=1}^M \left[ \frac{\pi_\theta(\mathcal{G}_m \mid \mathbf{s})}{\pi_{\overline{\theta}}(\mathcal{G}_m \mid \mathbf{s})} R(\mathcal{G}_m) \right] \right] \\
&= \frac{1}{M} \mathrm{Var}_{\mathcal{G} \sim \pi_{\overline{\theta}}(\cdot \mid \mathbf{s})} \left[ \rho_\theta(\mathcal{G} \mid \mathbf{s}) R(\mathcal{G}) \right].
\end{aligned}
\tag{19}
$$

Here, we have

$$
\begin{aligned}
&\mathrm{Var}_{\mathcal{G} \sim \pi_{\overline{\theta}}(\cdot \mid \mathbf{s})} \left[ \rho_\theta(\mathcal{G} \mid \mathbf{s}) R(\mathcal{G}) \right] \\
&= \mathbb{E}_{\mathcal{G} \sim \pi_{\overline{\theta}}(\cdot \mid \mathbf{s})} \left[ (\rho_\theta(\mathcal{G} \mid \mathbf{s}) R(\mathcal{G}))^2 \right] - \mathbb{E}_{\mathcal{G} \sim \pi_{\overline{\theta}}(\cdot \mid \mathbf{s})} \left[ \rho_\theta(\mathcal{G} \mid \mathbf{s}) R(\mathcal{G}) \right]^2 \\
&= \mathbb{E}_{\mathcal{G} \sim \pi_{\overline{\theta}}(\cdot \mid \mathbf{s})} \left[ (\rho_\theta(\mathcal{G} \mid \mathbf{s}) R(\mathcal{G}))^2 \right] - J^2(\theta),
\end{aligned}
\tag{20}
$$

and

$$
\begin{aligned}
\mathbb{E}_{\mathcal{G} \sim \pi_{\overline{\theta}}(\cdot \mid \mathbf{s})} \left[ (\rho_\theta(\mathcal{G} \mid \mathbf{s}) R(\mathcal{G}))^2 \right] &\le \mathbb{E}_{\mathcal{G} \sim \pi_{\overline{\theta}}(\cdot \mid \mathbf{s})} \left[ \rho_\theta^2(\mathcal{G} \mid \mathbf{s}) R_m^2 \right] \\
&= R_m^2 \mathbb{E}_{\mathcal{G} \sim \pi_{\overline{\theta}}(\cdot \mid \mathbf{s})} \left[ \rho_\theta^2(\mathcal{G} \mid \mathbf{s}) \right].
\end{aligned}
\tag{21}
$$

Notice that

$$
\begin{aligned}
\chi^2\Big(\pi_\theta(\cdot \mid \mathbf{s})\big\|\pi_{\overline{\theta}}(\cdot \mid \mathbf{s})\Big) &= \sum_{\mathcal{G}\in\mathbb{G}} \left(\frac{\pi_\theta(\mathcal{G}\mid\mathbf{s})}{\pi_{\overline{\theta}}(\mathcal{G}\mid\mathbf{s})} - 1\right)^2 \pi_{\overline{\theta}}(\mathcal{G}) \\
&= \sum_{\mathcal{G}\in\mathbb{G}} \left(1 + \frac{\pi_\theta^2(\mathcal{G}\mid\mathbf{s})}{\pi_{\overline{\theta}}^2(\mathcal{G}\mid\mathbf{s})} - 2\frac{\pi_\theta(\mathcal{G}\mid\mathbf{s})}{\pi_{\overline{\theta}}(\mathcal{G}\mid\mathbf{s})}\right) \pi_{\overline{\theta}}(\mathcal{G}\mid\mathbf{s}) \\
&= \sum_{\mathcal{G}\in\mathbb{G}} \pi_{\overline{\theta}}(\mathcal{G}\mid\mathbf{s}) - 2\sum_{\mathcal{G}\in\mathbb{G}} \pi_\theta(\mathcal{G}\mid\mathbf{s}) + \sum_{\mathcal{G}\in\mathbb{G}} \left(\rho_\theta^2(\mathcal{G}\mid\mathbf{s})\pi_{\overline{\theta}}(\mathcal{G}\mid\mathbf{s})\right) \\
&= \mathbb{E}_{\mathcal{G}\sim\pi_{\overline{\theta}}(\cdot\mid\mathbf{s})}\left[\rho_\theta^2(\mathcal{G}\mid\mathbf{s})\right] - 1. \quad\quad (22)
\end{aligned}
$$

Consequently, we have

$$
\mathrm{Var}_{\pi_{\overline{\theta}}(\cdot\mid\mathbf{s})}(\tilde{J}(\theta\mid\mathbf{s})) \le \frac{R_m^2}{M}\left(1 + \chi^2\Big(\pi_\theta(\cdot\mid\mathbf{s})\big\|\pi_{\overline{\theta}}(\cdot\mid\mathbf{s})\Big)\right) - \frac{J^2(\theta\mid\mathbf{s})}{M}. \quad\quad (23)
$$

$\square$

Proposition 1 indicates that the total variance is bounded by the $\chi^2$ divergence between policy $\pi_\theta(\cdot \mid \mathbf{s})$ and $\pi_{\overline{\theta}}(\cdot \mid \mathbf{s})$. Therefore, clipping the probability ratio reduces the variance of the policy gradient, thus enhancing training stability. This is especially necessary in discrete structural settings, where large updates can easily destabilize learning.

### C.3.7  Learning Setup

In the multi-relational graph encoder, we convert the directed graph into an undirected one and add self-loops for a bidirectional information flow, which is widely used in graph-structured data processing [19, 55]. The rationale behind this design lies in the fact that the directional information is sound and completely contained in node features. The node features used for the graph neural network initialization are provided in Table 3. It is worth noting that all node features are intrinsic to the node itself. For a fair comparison, the iterations of all experiments are set equal. We provide the hyperparameters of the main evaluations and ablation experiments in Table 4.

Table 3: Node Features

| Feature Description | Size |
| --- | --- |
| Column number | 1 |
| Stage number | 1 |
| One-hot encoding for node types, including 3:2 compressor, 2:2 compressor, initial pp, visual node | 4 |
| Sequential ID for compressor with the same column, stage, and type | 1 |

### C.3.8  Evaluation Metrics

The design of efficient multipliers involves optimizing multiple competing goals, including area and speed. To assess performance, we employ two key measures. Initially, we graphically compare the estimated Pareto frontiers for area versus latency across our approach and existing techniques. Additionally, we evaluate the hypervolume metric for these frontiers. Further details on these criteria are provided below.

**Multi-Objective Optimization Metrics**

Generally, for a multi-objective maximization task with n targets, the goal is identifying the optimal solution collection called the Pareto set. In such problems, solution x is said to dominate y if x performs equally or better in every objective while surpassing y in at least one, i.e., $\forall i \in [1, n], f_i(x) \ge f_i(y) \wedge \exists i \in [1, n], f_i(x) > f_i(y)$. A Pareto-optimal point remains undominated, and all such points form the Pareto set. Hypervolume, depicted in Figure 8, quantifies set

Table 4: Hyperparameters

| Hyperparameter | Value |
| --- | --- |
| Learning rate | 1e-4 |
| Iterations | 5000 |
| GNN layers | 3 |
| Hidden state dimensions | 512 |
| Query/key dimensions | 64 |
| Activation function | tanh |
| Optimizer | Adam [35] |
| Discretization penalty and output-port exclusivity constraint $\eta$ | 0.01 |
| PPO trust region $\epsilon$ | 0.2 |
| Elite pool size | 20 |

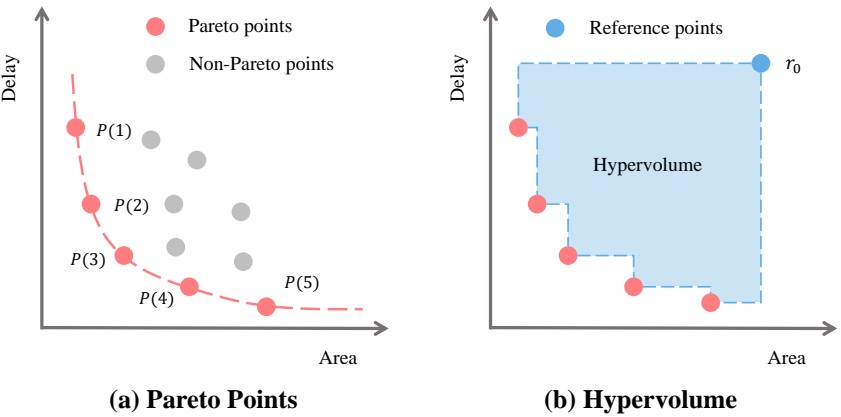

**(a) Pareto Points**       **(b) Hypervolume**

Figure 8: **(a)** An example for a Pareto optimal set with 2 objectives and 5 Pareto optimal solutions (Pareto points). **(b)** An example for hypervolume with a selected reference point $r_0$. Integrated area $H(P, r_0)$ is the union of the rectangular areas where the reference point $r_0$ and the Pareto point $P(i)$ are diagonally opposite corners.

quality by measuring dominated space volume. Computing hypervolume requires selecting a reference point; with fixed references, larger hypervolumes indicate superior Pareto sets.

**Definition 2** (Hypervolume metric). *Consider a set $P$ representing an approximate Pareto frontier in an $n$-dimensional optimization space, consisting of $N$ candidate points. Given a reference vector $r_0 \in R^m$, the hypervolume measure $\mathcal{H}(P, r_0)$ is formally expressed as:*

$$\mathcal{H}(P, r_0) = \int_{R^n} \mathbf{1}_{H(P,r_0)(z)dz}$$

*where $H(P, r_0) = \{z \in Z | \exists 1 \leq i \leq |P| : r_0 \preceq z \preceq P(i)\}$. Here, $P(i)$ denotes the $i$-th entry in $P$, $\preceq$ indicates Pareto dominance between vectors, and $\mathbf{1}_{H(P,r_0)}$ is an indicator function returning 1 when $z \in H(P, r_0)$ and 0 otherwise.*

### C.4 Implementation Details of Baselines

**GOMIL**[20] is a global optimization method that simultaneously considers the CT and CPA. The authors provide a C++ implementation. All coefficients related to the EDA toolchain and technology library are calibrated under the experimental setup. The resulting mixed integer programming (MIP) problem is solved using the Gurobi optimizer [75].

**UFO-MAC**[10] is a unified optimization CT and CPA framework considering compressor interconnections. The resulting integer programming problem is solved using the Gurobi optimizer [75].

**DOMAC**[11] is a recent differentiable compressor interconnection optimization algorithm, which optimizes a proxy delay model through routing permutation matrix relaxing.

**ArithTreeRL**[9] is a reinforcement learning method for CT optimization. The authors provide a Python implementation for AND PPG multipliers, which can be easily extended to MACs and Booth PPG multipliers.

**MUTE**[7] is the state-of-the-art (SOTA) hybrid RL-evolution CT optimization framework. The authors provide a Python implementation for multipliers and MACs.

# D More Experiment Results

## D.1 More Results of Main Evaluation

### D.1.1 More Results of Multiplier Evaluation

In this section, we provide our evaluations on Multiplie architectures, rencompassing 8-bit, 16-bit, 32-bit, and 64-bit implementations employing both AND gate-based and Booth encoding-based techniques. The min delay and min area are provided in Table 5.

Table 5: Min area and min delay of multiplier design tasks

| Mult.(AND) | 8 bit | | | | 16 bit | | | |
|---|---|---|---|---|---|---|---|---|
| **Method** | **Area($\mu m^2$) $\downarrow$** | **Impr.(%) $\uparrow$** | **Delay ($ns$) $\downarrow$** | **Impr. (%)$\uparrow$** | **Area($\mu m^2$) $\downarrow$** | **Impr.(%) $\uparrow$** | **Delay ($ns$) $\downarrow$** | **Impr.(%) $\uparrow$** |
| Wallace | 402 | NA | 0.74 | NA | 1,787 | NA | 1.37 | NA |
| GOMIL | 385 | 4.23 | 0.69 | 6.62 | 1,682 | 5.88 | 1.2 | 12.6 |
| UFO-MAC | 383 | **4.73** | 0.69 | 7.05 | 1,674 | **6.32** | 1.19 | 13.15 |
| DOMAC | 402 | 0.00 | 0.72 | 1.82 | 1,788 | -0.06 | 1.27 | 7.05 |
| ArithTreeRL | 385 | 4.23 | 0.7 | 4.37 | 1,707 | 4.48 | 1.12 | 18.38 |
| MUTE | 388 | 3.48 | 0.67 | 8.7 | 1,694 | 5.2 | 1.15 | 16.08 |
| **ARITH-DAS (Ours)** | 384 | 4.48 | 0.64 | **13.15** | 1,713 | 4.14 | 1.06 | **22.15** |
| Mult.(AND) | 32 bit | | | | 64 bit | | | |
| Method | **Area ($\mu m^2$) $\downarrow$** | **Impr.(%) $\uparrow$** | **Delay ($ns$) $\downarrow$** | **Impr.(%) $\uparrow$** | **Area ($\mu m^2$) $\downarrow$** | **Impr.(%) $\uparrow$** | **Delay ($ns$) $\downarrow$** | **Impr.(%) $\uparrow$** |
| Wallace | 7,402 | NA | 2.18 | NA | 29,721 | NA | 2.93 | NA |
| GOMIL | 7,029 | 5.04 | 2.06 | 5.21 | 28,602 | 3.77 | 2.66 | 9.33 |
| UFO-MAC | 7,027 | **5.07** | 2.11 | 2.96 | 28,600 | **3.77** | 2.72 | 7.17 |
| DOMAC | 7,489 | -1.18 | 1.99 | 8.5 | 30,055 | -1.12 | 2.49 | 15.01 |
| ArithTreeRL | 7,176 | 3.05 | 1.51 | 30.67 | 29,057 | 2.23 | 2.2 | 24.91 |
| MUTE | 7,125 | 3.74 | 1.64 | 24.49 | 28,677 | 3.51 | 2.07 | 29.56 |
| **ARITH-DAS (Ours)** | 7,099 | 4.09 | 1.41 | **35.35** | 28,669 | 3.54 | 1.85 | **36.94** |
| Mult.(Booth) | 8 bit | | | | 16 bit | | | |
| **Method** | **Area ($\mu m^2$) $\downarrow$** | **Impr.(%) $\uparrow$** | **Delay ($ns$) $\downarrow$** | **Impr.(%) $\uparrow$** | **Area ($\mu m^2$) $\downarrow$** | **Impr.(%) $\uparrow$** | **Delay ($ns$) $\downarrow$** | **Impr.(%) $\uparrow$** |
| Wallace | 408 | NA | 0.76 | NA | 1,505 | NA | 1.32 | NA |
| GOMIL | 406 | 0.49 | 0.68 | 10.09 | 1,407 | 6.51 | 1.16 | 12.50 |
| UFO-MAC | 405 | 0.74 | 0.69 | 8.46 | 1,406 | **6.58** | 1.2 | 9.25 |
| DOMAC | 415 | -1.72 | 0.74 | 2.07 | 1,548 | -2.86 | 1.27 | 3.59 |
| ArithTreeRL | 405 | 0.74 | 0.71 | 6.36 | 1,425 | 5.32 | 1.07 | 18.73 |
| MUTE | 406 | 0.49 | 0.68 | 10.55 | 1,432 | 4.85 | 1.05 | 20.53 |
| **ARITH-DAS (Ours)** | 403 | **1.23** | 0.64 | **15.91** | 1,418 | 5.78 | 1.01 | **23.64** |
| Mult.(Booth) | 32 bit | | | | 64 bit | | | |
| **Method** | **Area ($\mu m^2$) $\downarrow$** | **Impr.(%) $\uparrow$** | **Delay ($ns$) $\downarrow$** | **Impr.(%) $\uparrow$** | **Area ($\mu m^2$) $\downarrow$** | **Impr.(%) $\uparrow$** | **Delay ($ns$) $\downarrow$** | **Impr.(%) $\uparrow$** |
| Wallace | 5,383 | NA | 2.24 | NA | 20,435 | NA | 2.94 | NA |
| GOMIL | 5,176 | 3.85 | 1.94 | 13.51 | 19,677 | 3.71 | 2.66 | 9.46 |
| UFO-MAC | 5,175 | **3.86** | 1.97 | 12.42 | 19,675 | **3.72** | 2.82 | 4.04 |
| DOMAC | 5,500 | -2.17 | 1.87 | 16.79 | 20,787 | -1.72 | 2.22 | 24.41 |
| ArithTreeRL | 5,227 | 2.9 | 1.68 | 25.33 | 19,890 | 2.67 | 2.04 | 30.48 |
| MUTE | 5,328 | 1.02 | 1.49 | 33.6 | 19,856 | 2.83 | 2.03 | 31.01 |
| **ARITH-DAS (Ours)** | 5,210 | 3.21 | 1.44 | **35.78** | 19,690 | 3.65 | 1.92 | **34.53** |

## D.1.2 More Results of MAC Evaluation

We provide our evaluations on MACs in this section. The visualization of the Pareto front is illustrated in Figure 9. The hypervolume is recorded in Table 6, and the min delay and min area are provided in Table 7.

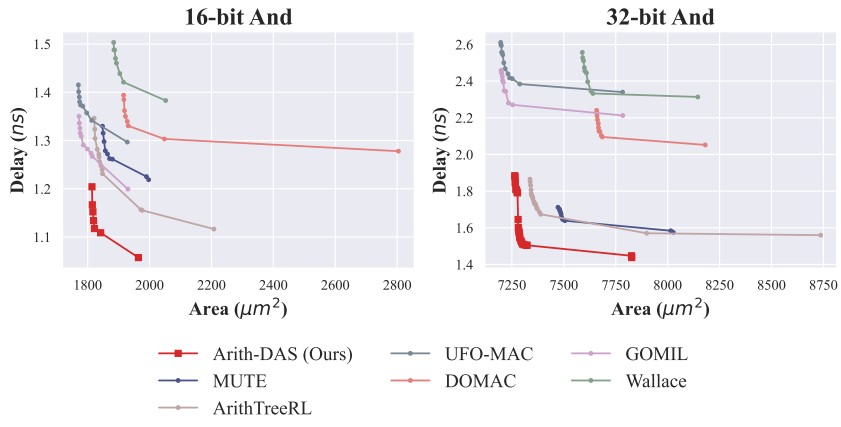

Figure 9: Pareto front of MAC designs.

Table 6: Hypervolumn of MACs

| MACs | 16 bit | | 32 bit | |
|---|---|---|---|---|
| Method | Hypervolume↑ | Impr.(%) | Hypervolume↑ | Impr.(%) |
| Wallace | 103.17 | NA | 325.72 | NA |
| GOMIL | 301.78 | 192.51 | 575.14 | 76.57 |
| UFO-MAC | 205.20 | 98.89 | 385.01 | 18.20 |
| DOMAC | 173.67 | 68.33 | 579.01 | 77.76 |
| ArithTreeRL | 352.35 | 241.52 | 1,395.40 | 328.40 |
| MUTE | 265.24 | 157.08 | 1,274.29 | 291.22 |
| **ARITH-DAS (Ours)** | **433.18** | **319.87** | **1,684.68** | **417.21** |

### D.1.3 More Results of Generalization Evaluation

We provide the Hypervolume of PE arrays in Table 8. The min delays and min areas are provided in Table 9. The area and delay of Koios benchmark [39] designs integrated with our ARITH-DAS and baselines are provided in Table 10.

### D.2 Visualization of Results

To provide an intuitive understanding of the problem, we visualize representative examples of the circuit structure. Due to the large problem scale, we only present the complete results for 8-bit multipliers using AND and Booth encoding and 16-bit multipliers using AND encoding, as shown in Figure 11, Figure 10, and Figure 12. Additionally, for the 32-bit and 64-bit AND multiplier, we selectively visualize Columns 30-31 and 32-33 to highlight the structural patterns in deeper regions of the compression tree, as shown in Figure 13, Figure 14, Figure 15 and Figure 16.
In all figures:

- Green nodes represent virtual nodes, typically containing 1 input and 1 or 0 outputs.

Table 7: Min area and min delay of MAC designs

| MAC | 16 bit | | | | 32 bit | | | |
|---|---|---|---|---|---|---|---|---|
| Method | Area $(\mu m^2)\downarrow$ | Impr.(%)↑ | Delay(ns)↓ | Impr.(%)↑ | Area $(\mu m^2)\downarrow$ | Impr.(%)↑ | Delay(ns)↓ | Impr.(%)↑ |
| Wallace | 1,884 | NA | 1.38 | NA | 7,589 | NA | 2.31 | NA |
| GOMIL | 1,772 | 5.94 | 1.20 | 13.28 | 7,198 | 5.15 | 2.21 | 4.37 |
| UFO-MAC | 1,770 | **6.05** | 1.30 | 6.23 | 7,196 | **5.18** | 2.34 | -1.14 |
| DOMAC | 1,916 | -1.70 | 1.28 | 7.59 | 7,658 | -0.91 | 2.05 | 11.31 |
| ArithTreeRL | 1,821 | 3.34 | 1.12 | 19.26 | 7,337 | 3.32 | 1.56 | 32.59 |
| MUTE | 1,848 | 1.91 | 1.22 | 11.87 | 7,472 | 1.54 | 1.58 | 31.91 |
| **ARITH-DAS (Ours)** | 1,814 | 3.72 | 1.06 | **23.54** | 7,264 | 4.28 | 1.44 | **37.87** |

Table 8: Hypervolume of PE arrays.

| PE Array | 16-bit AND | | 16-bit Booth | | 32-bit AND | | 32-bit Booth | |
|---|---|---|---|---|---|---|---|---|
| **Method** | **Hypervolume ↑** | **Impr.(%) ↑** | **Hypervolume ↑** | **Impr. (%)↑** | **Hypervolume ↑** | **Impr.(%) ↑** | **Hypervolume ↑** | **Impr.(%) ↑** |
| Wallace | 13,023.58 | NA | 12,493.99 | NA | 64,431.31 | NA | 57,036.10 | NA |
| GOMIL | 3,774.34 | -71.02 | 18,798.77 | 50.46 | 84,133.59 | 30.58 | 120,763.99 | 111.73 |
| UFO-MAC | 5,393.81 | -58.58 | 15,179.04 | 21.49 | 76,563.34 | 18.83 | 111,945.32 | 96.27 |
| DOMAC | 12,865.94 | -1.21 | 11,417.57 | -8.62 | 100,289.37 | 55.65 | 99,131.37 | 73.80 |
| ArithTreeRL | 15,767.41 | 21.07 | 14,934.67 | 19.53 | 133,033.11 | 106.47 | 116,160.12 | 103.66 |
| MUTE | 16,160.54 | 24.09 | 14,330.01 | 14.70 | 129,010.95 | 100.23 | 151,738.71 | 166.04 |
| **ARITH-DAS (Ours)** | 16,219.91 | **24.54** | 20,680.92 | **65.53** | 152,373.47 | **136.49** | 194,991.71 | **241.87** |

Table 9: Min area and min delay of PE arrays

| PE array | 16-bit AND | | | | 16-bit Booth | | | |
|---|---|---|---|---|---|---|---|---|
| **Method** | **Area($\mu m^2$) ↓** | **Impr.(%) ↑** | **Delay($ns$) ↓** | **Impr. (%)↑** | **Area($\mu m^2$) ↓** | **Impr.(%) ↑** | **Delay($ns$) ↓** | **Impr.(%) ↑** |
| Wallace | 149,556 | NA | 1.72 | NA | 131,459 | NA | 1.78 | NA |
| GOMIL | 142,729 | 4.56 | 1.81 | -5.23 | 125,126 | 4.82 | 1.7 | 4.49 |
| UFO-MAC | 142,218 | **4.91** | 1.82 | -5.81 | 125,024 | **4.9** | 1.7 | 4.49 |
| DOMAC | 149,522 | 0.02 | 1.68 | 2.33 | 131,136 | 0.25 | 1.79 | -0.56 |
| ArithTreeRL | 144,329 | 3.5 | 1.64 | 4.65 | 125,927 | 4.21 | 1.74 | 2.25 |
| MUTE | 143,359 | 4.14 | 1.67 | 2.91 | 125,637 | 4.43 | 1.77 | 0.56 |
| **ARITH-DAS (Ours)** | 145,249 | 2.88 | 1.62 | **5.81** | 126,710 | 3.61 | 1.67 | **6.18** |

| PE array | 32-bit AND | | | | 32-bit Booth | | | |
|---|---|---|---|---|---|---|---|---|
| Method | **Area($\mu m^2$) ↓** | **Impr.(%) ↑** | **Delay($ns$) ↓** | **Impr.(%) ↑** | **Area($\mu m^2$) ↓** | **Impr.(%) ↑** | **Delay($ns$) ↓** | **Impr. (%)↑** |
| Wallace | 545,126 | NA | 2.73 | NA | 417,650 | NA | 2.78 | NA |
| GOMIL | 521,445 | 4.34 | 2.62 | 4.03 | 404,456 | 3.16 | 2.49 | 10.43 |
| UFO-MAC | 521,309 | **4.37** | 2.67 | 2.2 | 404,354 | **3.18** | 2.5 | 10.07 |
| DOMAC | 551,526 | -1.17 | 2.56 | 6.23 | 425,191 | -1.81 | 2.55 | 8.27 |
| ArithTreeRL | 531,540 | 2.49 | 2.4 | 12.09 | 407,691 | 2.38 | 2.47 | 11.15 |
| MUTE | 527,948 | 3.15 | 2.4 | 12.09 | 413,836 | 0.91 | 2.29 | 17.63 |
| **ARITH-DAS (Ours)** | 526,348 | 3.44 | 2.34 | **14.29** | 407,384 | 2.46 | 2.13 | **23.38** |

- Blue nodes represent half-adder nodes, typically containing 2 inputs and 2 outputs.

- Orange nodes represent full-adder nodes, typically containing 3 inputs and 2 outputs.

- Gray nodes in the first row represent partial product nodes, typically containing 1 output. Gray nodes in the last row represent output nodes.

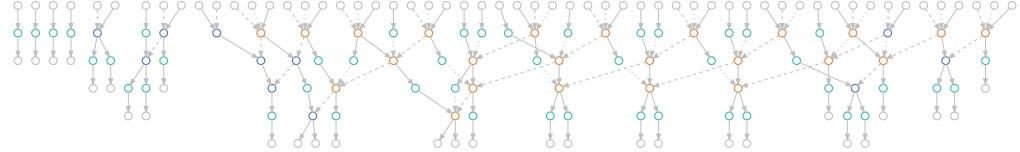

Figure 10: 8-bit Booth Multiplier

## D.3 Runtime Comparison

This section compares the runtime of our method with baseline approaches under consistent experimental settings. Specifically, we run all post-synthesis driven algorithms for 5000 iterations. For programming-based methods, we report the objective gap with 1 hour runtime, which is defined by (Incumbent − BestBd)/Incumbent × 100% [75]. For the proxy-delay-model driven method, DO-MAC [11], we evaluate it with 1000 iterations. Although our method does not outperform traditional solvers in terms of runtime, it offers a more flexible and generalizable optimization framework that is well-suited for large-scale and complex circuit design problems. The detailed runtime environment is provided in Appendix C.1 and Appendix C.2. The runtime results are summarized in Table 11.

## E Licenses

We credit the following open-source project used in this paper. Our main codes are provided at
`https://github.com/dakfjalka/Arith-DAS.git`.

Table 10: Area and Delay of Koios benchmark integrated with our ARITH-DAS and baselines.

| Design | Method | Area($\mu m^2$)↓ | Area Impr.(%) ↑ | Delay($ns$)↓ | Delay Impr.(%) ↑ |
|---|---|---|---|---|---|
| attention_layer | Wallace | 1.6590E+06 | NA | 1.55 | NA |
| | GOMIL | 1.6483E+06 | 0.64 | 1.42 | 8.39 |
| | DOMAC | 1.6579E+06 | 0.06 | 1.56 | -0.65 |
| | UFO-MAC | 1.6481E+06 | **0.65** | 1.56 | -0.65 |
| | ArithTreeRL | 1.6487E+06 | 0.62 | 1.35 | 12.9 |
| | MUTE | 1.6529E+06 | 0.37 | 1.37 | 11.61 |
| | **ARITH-DAS (Ours)** | 1.6495E+06 | 0.57 | 1.29 | **16.77** |

| Design | Method | Area($\mu m^2$)↓ | Area Impr.(%) ↑ | Delay($ns$)↓ | Delay Impr.(%) ↑ |
|---|---|---|---|---|---|
| conv_layer | Wallace | 1.0407E+07 | NA | 1.74 | NA |
| | GOMIL | 1.0400E+07 | 0.07 | 1.63 | 6.32 |
| | DOMAC | 1.0404E+07 | 0.02 | 1.76 | -1.15 |
| | UFO-MAC | 1.0400E+07 | 0.07 | 1.76 | -1.15 |
| | ArithTreeRL | 1.0402E+07 | 0.04 | 1.52 | 12.64 |
| | MUTE | 1.0402E+07 | 0.05 | 1.47 | 15.52 |
| | **ARITH-DAS (Ours)** | 1.0398E+07 | **0.08** | 1.41 | **18.97** |

| Design | Method | Area($\mu m^2$)↓ | Area Impr.(%) ↑ | Delay($ns$)↓ | Delay Impr.(%) ↑ |
|---|---|---|---|---|---|
| eltwise_layer | Wallace | 1.3207E+07 | NA | 0.97 | NA |
| | GOMIL | 1.3207E+07 | 0 | 0.9 | 7.22 |
| | DOMAC | 1.3153E+07 | 0.41 | 0.95 | 2.06 |
| | UFO-MAC | 1.3207E+07 | 0 | 0.96 | 1.03 |
| | ArithTreeRL | 1.3234E+07 | -0.2 | 0.91 | 6.19 |
| | MUTE | 1.3234E+07 | -0.2 | 0.94 | 3.09 |
| | **ARITH-DAS (Ours)** | 1.3145E+07 | **0.47** | 0.86 | **11.34** |

| Design | Method | Area($\mu m^2$)↓ | Area Impr.(%) ↑ | Delay($ns$)↓ | Delay Impr.(%) ↑ |
|---|---|---|---|---|---|
| gemm_layer | Wallace | 2.1045E+06 | NA | 0.97 | NA |
| | GOMIL | 2.0977E+06 | 0.32 | 0.9 | 7.22 |
| | DOMAC | 2.1045E+06 | 0 | 0.95 | 2.06 |
| | UFO-MAC | 2.0968E+06 | **0.36** | 0.95 | 2.06 |
| | ArithTreeRL | 2.0977E+06 | 0.32 | 0.91 | 6.19 |
| | MUTE | 2.0986E+06 | 0.28 | 0.94 | 3.09 |
| | **ARITH-DAS (Ours)** | 2.0984E+06 | 0.29 | 0.86 | **11.34** |

| Design | Method | Area($\mu m^2$)↓ | Area Impr.(%) ↑ | Delay($ns$)↓ | Delay Impr.(%) ↑ |
|---|---|---|---|---|---|
| lstm | Wallace | 7.5409E+06 | NA | 1.57 | NA |
| | GOMIL | 7.4785E+06 | 0.83 | 1.38 | 12.1 |
| | DOMAC | 7.5390E+06 | 0.03 | 1.56 | 0.64 |
| | UFO-MAC | 7.4775E+06 | **0.84** | 1.43 | 8.92 |
| | ArithTreeRL | 7.4885E+06 | 0.7 | 1.35 | 14.01 |
| | MUTE | 7.5078E+06 | 0.44 | 1.37 | 12.74 |
| | **ARITH-DAS (Ours)** | 7.4927E+06 | 0.64 | 1.29 | **17.83** |

| Design | Method | Area($\mu m^2$)↓ | Area Impr.(%) ↑ | Delay($ns$)↓ | Delay Impr.(%) ↑ |
|---|---|---|---|---|---|
| softmax | Wallace | 1.2948E+05 | NA | 0.97 | NA |
| | GOMIL | 1.2921E+05 | 0.21 | 0.91 | 6.19 |
| | DOMAC | 1.2948E+05 | 0 | 0.95 | 2.06 |
| | UFO-MAC | 1.2917E+05 | **0.24** | 0.95 | 2.06 |
| | ArithTreeRL | 1.2921E+05 | 0.21 | 0.91 | 6.19 |
| | MUTE | 1.2924E+05 | 0.18 | 0.94 | 3.09 |
| | **ARITH-DAS (Ours)** | 1.2924E+05 | 0.19 | 0.85 | **12.37** |

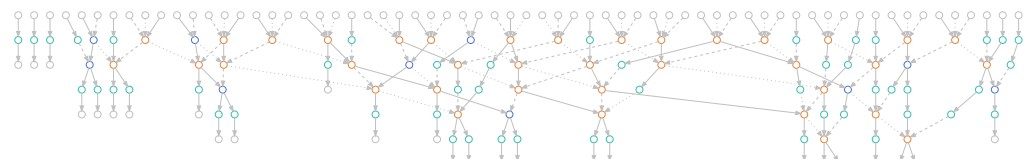

Figure 11: 8-bit AND Multiplier

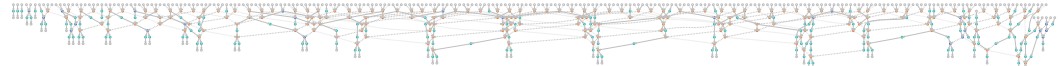

Figure 12: 16-bit AND Multiplier

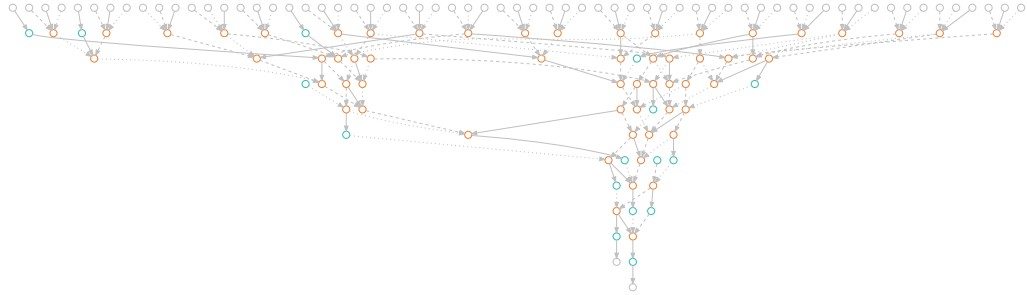

Figure 13: 32-bit AND Multiplier (Column 30 and Column 31)

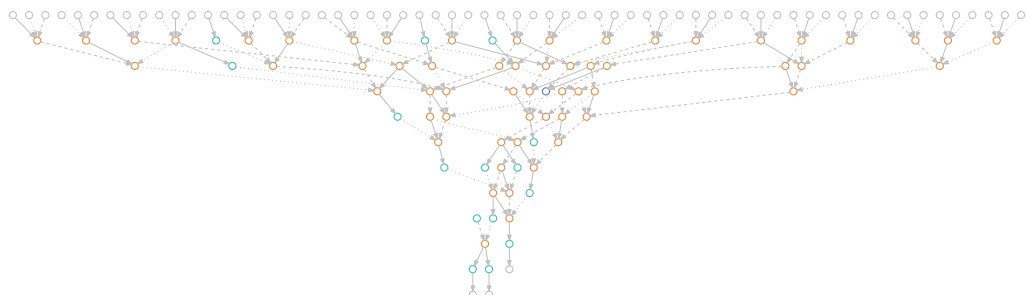

Figure 14: 32-bit AND Multiplier (Column 31 and Column 32)

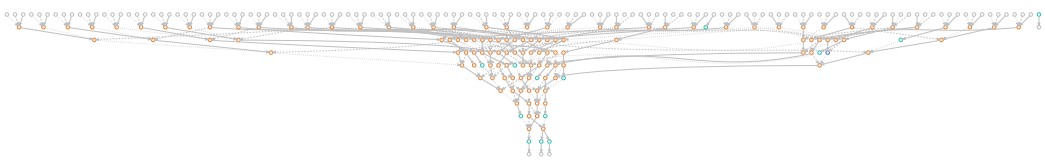

Figure 15: 64-bit AND Multiplier (Column 62 and Column 63)

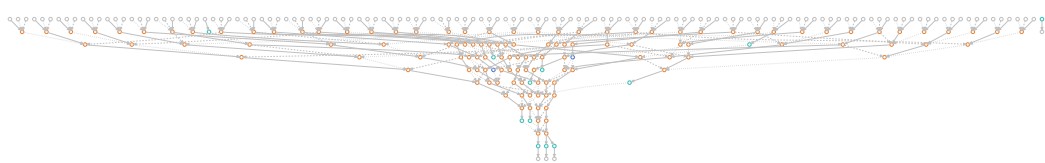

Figure 16: 64-bit AND Multiplier (Column 63 and Column 64)

Table 11: Runtime Comparison

| Medthod | 8-bit AND Mult. | 16-bit AND Mult. | 32-bit AND Mult. |
|---|---|---|---|
| Wallace | NA | NA | NA |
| Programing based algorithms objective gap with 1 hour runtime (%) | | | |
| GOMIL | 0% (solved in 0.51 sec.) | 0.08% | 0.59% |
| UFO-MAC | 0% (solved in 2.46 sec.) | 0% (solved in 70.35 sec.) | 33.0% |
| Proxy-delay-model driven algorithms runtime for 1000 iterations (**minute**) | | | |
| DOMAC | 1.50 | 5.46 | 19.42 |
| Post-synthesis driven algorithms runtime for 5000 iterations (**hour**) | | | |
| ArithTreeRL | 5.46 | 10.42 | 25.89 |
| MUTE | 9.37 | 12.60 | 50.83 |
| **ARITH-DAS (Ours)** | 6.40 | 13.06 | 38.73 |

1. OpenROAD [29]: BSD 3-Clause License

2. Yosys [30]: ISC License

3. OpenSTA [69]: GPL-3.0 License

4. Verilator [32]: LGPL-3.0 License

5. ArithTreeRL [9]: No License

6. GOMIL [20]: No License

7. VTR/Koios [39]: VTR License

