# OpenReview forum: "High-Performance Arithmetic Circuit Optimization via Differentiable Architecture Search"
_NeurIPS.cc/2025/Conference — NeurIPS 2025 spotlight_

### Official Review · Reviewer_tJhd · 2025-06-30

**Clarity:** 2
**Significance:** 3
**Originality:** 2
**Rating:** 4
**Confidence:** 2

**Summary:**

Arithmetic circuit design is the basis of modern hardware platforms. A lot of works have aimed to design better arithmetic circuits, and they have focused on coarse-grained designs (compressor allocation in compressor tree). In the meantime, those works have relied on heuristics to design fine-grained parts.

This work proposes to optimize compressor interconnections, which is a fine-grained design of arithmetic circuits. It denotes compressor interconnection architecture as a graph, regarding compressors as nodes and connections between any two compressors as edges. By doing so, compressor interconnection becomes a link prediction problem that can be solved differentially with GNN and graph attention.

However, there is a critical barrier to optimizing compressor interconnections with GNN. The search space of compressor interconnections is discrete, thus it is not suitable for solving the problem with neural networks. To resolve this issue, this work relaxes the discrete search space into continuous search space by denoting edges as softmax-based soft labels, instead of hard labels.

With the above proposals, the paper achieves Pareto-dominance results than other prior works.

**Questions:**

It seems that the compressor tree search part is based on MUTE. However, in the experiments in the appendix, the area of the circuits designed by the proposal is smaller than that of MUTE. Is there any difference between the two methods while searching compressor trees, or does it come from the stochasticity of the evolution-based method?

**Ethical Concerns:**

["NO or VERY MINOR ethics concerns only"]

**Final Justification:**

With the authors’ feedback and other reviews comments, the reviewer keep the rating of acceptance.

**Limitations:**

Yes

**Paper Formatting Concerns:**

There is no formatting concerns

**Quality:**

3

**Strengths And Weaknesses:**

Strengths
- It is noticeable that propose to design arithmetic circuits differentiably based on GNN.
- This work optimizes arithmetic circuit design by focusing on interconnections, which haven’t been addressed.
- Search space relaxation tailored loss function for the problem leads to achieving state-of-the-art results for designing arithmetic circuits.
- The paper supports the claim well with mathematic formulas.

Weaknesses
- It seems that the paper’s Compressor Tree design, which previous L2O works have aimed, is highly rooted in the previous work (MUTE).
- If so, it would be better to demonstrate the effectiveness of the proposed interconnection design by integrating the proposal with other prior works.

---

> ### Author Rebuttal · Authors · 2025-07-30
>
> ## Response to Reviewer tJhd
>
> We sincerely thank you for your careful review and for your constructive, insightful, and valuable comments. Please find below our detailed responses to your inquiries.
>
> ### **Weakness 1 & Weakness 2: Integrating with Prior Works**
> >  **It would be better to demonstrate the effectiveness of the proposed interconnection design by integrating the proposal with other prior works.**
>
> Thank you for your valuable suggestion! In response, we have **integrated our proposed interconnection generation method into two representative learning-to-optimize arithmetic circuit optimization frameworks**: RL-MUL [1] and ArithTreeRL [2], and report the results in the table below.
>
> |  | 16bit-AND |  | 32bit-AND |  |
> |:---:|:---:|:---:|:---:|:---:|
> | Method | HV. | Impr. | HV. | Impr. |
> | ArithTreeRL | $315.19$ | NA | $2779.56$ | NA |
> | ArithTreeRL + Ours | $353.17$ | $+12.05\\%$ | $2811.92$ | $+1.16\\%$ |
> |  |  |  |  |  |
> | RL-MUL | $275.67$ | NA | $2600.24$ | NA |
> | RL-MUL + Ours | $309.07$ | $+12.11\\%$ | $2779.85$ | $+6.91\\%$ |
>
> **1. Why We Build Our Approach Upon MUTE**
>
> As noted in the main paper, our method ARITH-DAS is implemented atop the MUTE framework. This decision is motivated by two primary considerations.
>
> **First**, MUTE represents one of **the most recent and competitive State-of-the-Art (SOTA) methods** for arithmetic circuit optimization. It provides a solid and well-maintained baseline, with strong performance across various circuit configurations, making it a reliable foundation for developing and benchmarking our proposed interconnection strategy.
>
> **Second**, **the evolutionary nature of MUTE is particularly advantageous for our learning framework**. Evolutionary methods inherently explore **a broader and more diverse design space** compared to purely gradient-based or heuristic approaches. This **diversity of intermediate designs** generated during the evolution process enables the collection of a larger and more representative training set, which is crucial for effectively learning a **generalizable and robust** interconnection policy. As a result, MUTE not only provides a strong backbone for implementation but also naturally facilitates the data-driven learning required by our method.
>
> **2. Integrating Our Interconnection Approach with Prior Works**
>
> As demonstrated in the main paper, our method is **modular and compatible** with other learning-to-optimize frameworks as well. Specifically, we integrate our pre-trained model and corresponding interconnection generation strategy into **the full Verilog synthesis pipeline** of both RL-MUL and ArithTreeRL. RL-MUL is a Deep Q-Network (DQN)-based method that sequentially refines a pre-defined compression tree structure, such as a Wallace Tree, by making local decisions. In contrast, ArithTreeRL employs a Proxy-Policy Gradient approach to construct the entire compressor tree from scratch. **In both methods, the original handcrafted sequential interconnection policy is replaced by our *pre-trained* interconnection generator**. Notably, ArithTreeRL is already included as a baseline in our main paper.
>
> To ensure fairness and tractability given limited compute resources, we evaluate both original frameworks and their integrated versions on two widely used multipliers, 16-bit and 32-bit AND multipliers. The results in the table above show that our interconnection design **substantially improves the hypervolume (HV) for 16-bit designs, and notably enhances the 32-bit designs as well**.
>
> We attribute the larger gains in the 16-bit case to its smaller design space, where interconnection quality plays a more dominant role. In contrast, the 32-bit design space is much broader, and the relative impact of interconnect optimization is partially diluted by other factors. Nevertheless, **the improvements are consistent across both scales**, validating the **generalizability and effectiveness** of our method.
>
> ---
>
> ### **Question 2: Area Comparison with MUTE**
> > **Is there any difference between the two methods while searching compressor trees, or does the difference of area come from the stochasticity of the evolution-based method?**
>
> **1. Reason for the Better Area Metric of ARITH-DAS**
>
> Our method leverages the MUTE framework as the compressor allocation search engine, and we do not alter its evolutionary search algorithm. The difference observed in the area metric is **partially attributable to the inherent stochasticity** of the evolution-based search, which relies on randomized mutation and selection operations. Small variations introduced at intermediate generations may propagate through the search process, ultimately leading to different final designs.
>
> However, **the major factor underlying the area difference lies in the objective setting**. Both our method and MUTE adopt the same bi-objective optimization formulation:
>
> $$
> \min_{\mathcal{G}\in\mathbb{G}} - R(\mathcal{G}) = w_1\cdot \operatorname{area}(\mathcal{G}) + w_2\cdot \operatorname{delay}(\mathcal{G}),
> $$
>
> where $\operatorname{area}(\mathcal{G})$ and $\operatorname{delay}(\mathcal{G})$ represent the synthesized area and delay of a compressor tree $\mathcal{G}$, and $w_1, w_2$ control the trade-off. Because the objective function is a weighted sum, **a design with excellent area but suboptimal delay may be discarded during the search if its overall weighted cost is not competitive**. In our method, the pre-trained interconnection generator performs targeted optimizations that directly improve the delay of the generated compressor trees. This delay improvement increases the likelihood that certain candidates are retained and refined in later generations. Over multiple iterations, this mechanism shifts the distribution of surviving designs and may result in final circuits that are superior in both delay and, indirectly, area metrics.
>
> Compared with other methods that directly optimize the area metric alone, such as GOMIL and UFO-MAC, **neither MUTE nor our ARITH-DAS is guaranteed to always favor the design with the smallest area**. Instead, designs that strike a **better trade-off between area and delay** tend to be preserved and further refined by the evolutionary search.
>
> **2. Numerical Example**
>
> To illustrate this more concretely, suppose $w_1=w_2=0.5$, and consider two candidate designs $A$ and $B$:
>
> - $\operatorname{area}(A) = 2, \operatorname{delay}(A) = 2, - R(A) = 2$.
> - $\operatorname{area}(B) = 1, \operatorname{delay}(B) = 4, - R(B) = 2.5$.
>
> Under the combined cost function, design $A$ would be the preferred candidate for both MUTE and ARITH-DAS, despite its slightly larger area, because its overall weighted cost is lower.
>
> However, ARITH-DAS integrates a delay-aware interconnection generator, and the delay performance of candidate designs is further improved. For example, for the same compressor tree allocation with design $B$, ARITH-DAS optimized  interconnection topology and produces a new design $B'$:
>
> - $\operatorname{area}(B') = 1, \operatorname{delay}(B') = 2, - R(A) = 1.5$.
>
> Now, the improved design $B'$ achieves a much lower combined cost than $A$. As a result, ARITH-DAS would preserve $B'$ in the evolutionary search, whereas MUTE would have previously discarded $B$ due to its high delay. Over multiple iterations, this mechanism enables ARITH-DAS to retain and refine higher-quality solutions that strike a better balance between area and delay, ultimately leading to superior final designs.
>
> ---
>
> ### Reference
>
> [1] Zuo D, Ouyang Y, Ma Y. Rl-mul: Multiplier design optimization with deep reinforcement learning[C]//2023 60th ACM/IEEE Design Automation Conference (DAC). IEEE, 2023: 1-6.
>
> [2] Lai Y, Liu J, Pan D Z, et al. Scalable and effective arithmetic tree generation for adder and multiplier designs[J]. Advances in Neural Information Processing Systems, 2024, 37: 139151-139178.

---

> > ### Author Response · Authors · 2025-08-08
> > **We are looking forward to your feedback.**
> >
> > Dear Reviewer tJhd,
> >
> > We are writing as the authors of the paper "High-Performance Arithmetic Circuit Optimization via Differentiable Architecture Search" (ID: 25851). We sincerely thank you for your time and efforts during the rebuttal process. The deadline for the author-reviewer discussion period is approaching. We are looking forward to your feedback to understand if our responses have adequately addressed your concerns. If so, we would be grateful for your acknowledgment that the revisions have sufficiently clarified and strengthened our work. If not, please let us know your further concerns, and we will continue actively responding to your comments. We sincerely thank you once more for your insightful comments and kind support.
> >
> > Best,
> >
> > Authors

---

### Official Review · Reviewer_CFYW · 2025-07-05

**Clarity:** 3
**Significance:** 4
**Originality:** 3
**Rating:** 5
**Confidence:** 2

**Summary:**

This paper aims to enhance the Arithmetic Circuit optimization via a differentiable architecture search framework. To do so, the author first model the combinational logic circuits as directed acyclic graphs (DAGs), in which partial products and compressors are modeled as nodes, and logical dependencies are modeled as directed edges. To capture this asymmetry, they represent the compressor tree as a multi-relational graph, where each edge type corresponds to a specific input port of the target node.  Then, the paper proposes a method called **ARITH-DAS**,encoding structural information via a multi-relational graph neural network and modeling interconnect prediction through graph attention, it enables end-to-end, gradient-based optimization directly guided by post-synthesis performance metrics. ARITH-DAS can also perform a proxy-free PPO-like training to optimize the interconnect structure with respect to post-synthesis performance metrics. The experiments shows ARITH-DAS's effectiveness and superiority when comparing it to other heuristic and ML-based baselines in area-delay trade-offs.

**Questions:**

See weaknesses.

**Ethical Concerns:**

["NO or VERY MINOR ethics concerns only"]

**Final Justification:**

During the rebuttal the authors provided more details and basically solved my concerns.

**Limitations:**

Yes

**Paper Formatting Concerns:**

No.

**Quality:**

3

**Strengths And Weaknesses:**

# Strengths
1. The research questions are very practical.
2. The paper is able to illustrate the concept of arithmetic circuit to readers who are not familiar with this topic
3. The proposed method is effective and the ablation study also states the importance of each proposed module.


# Weakness
1. Some details of ARITH-DAS are not fully explained. For example, in lines 168-181, the author mentions \mathbb{s} as the matrix of allocation of a compressor tree without demonstrating the role of \mathbb{s} in terms of DAG setup or rest of the method implementation.
2.  There is one paper: Reinforcement Learning for Combinatorial Optimization [1] which might be associate this submission, and the author may also include it as one reference.

[1] Darvariu, Victor-Alexandru, Stephen Hailes, and Mirco Musolesi. "Graph reinforcement learning for combinatorial optimization: A survey and unifying perspective." arXiv preprint arXiv:2404.06492 (2024).

---

> ### Author Rebuttal · Authors · 2025-07-30
>
> ## **Response to Reviewer CFYW**
>
> We sincerely thank you for your careful review and for your constructive, insightful, and valuable comments. Please find below our detailed responses to your inquiries.
>
> ### **Weakness 1: More Details About Compressor Allocation Matrix**
> > **Some details of ARITH-DAS are not fully explained. For example, the matrix of allocation $\mathbf{s}$ is mentioned without further demonstration.**
>
> Thank you for your valuable suggestion! We provide further demonstration of the compressor allocation matrix below, and **we will add a dedicated subsection in the revised version of our paper** to clarify its definition, role, and how it interacts with the rest of the framework.
>
> **1. Further Demonstration for Compressor Allocation Matrix $\mathbf{s}$ and Seach Space $\mathbb{S}$**
>
> For a compressor tree with bit-width $N$ and $T$ types of different compressors, where $N,T\in \mathbb{N} _+$, the allocation matrix $\mathbf{s}\in\mathbb{N} _ +^{T\times N}$ stands for the number of specific compressors allocated in the column:
>
> $$
> \mathbf{s} = \left[\begin{matrix}
>  s_{1,1} & s_{1,2} & \dots & s_{1,N} \\\\
>  s_{2,1} & s_{2,2} & \dots & s_{2,N} \\\\
>  \vdots & \vdots & \ddots & \vdots \\\\
>  s_{T,1} & s_{T,2} & \dots & s_{T,N} \\\\\
> \end{matrix}\right].
> $$
>
> Here, $s_{t,n}$ denotes the number of compressors of type $t$ allocated at bit column $n$. Different choices of $\mathbf{s}$ define different compressor tree configurations, and the collection of all valid allocation matrices forms the allocation-level search space $\mathbb{S} \subset \mathbb{N} _+^{T\times N}$.
>
> In our case and all the baselines, we employ two different types of compressors and therefore $T=2$. Without loss of generality, $t=1$ stands for 3:2 Compressor, and $t=2$ stands for 2:2 Compressor. Let $\mathbf{p}\in\mathbb{N}_+^{N}$ denote the partial product array, which is a fixed matrix when given input bit-width and encoding circuit. For example, $\mathbf{p} = [1, 2, 3, 4, 3, 2, 1]^\top$ for a 4-bit AND multiplier as illustrated in Figure 1 in the main paper. All allocation matrices $\mathbf{s}\in\mathbb{S}$ should satisfy the following constraints to ensure valid compressor tree reduction:
>
> $$
> \begin{cases}
> p_n - 2 s_{1,n} - s_{2,n} \ge 1 & n = 1\\\\
> p_n - 2 s_{1,n} - s_{2,n} + s_{1,n-1} + s_{2,n-1} \ge 1 & n > 1\\
> \end{cases},
> $$
>
> $$
> \begin{cases}
> p_n - 2 s_{1,n} - s_{2,n} \le 2 & n = 1\\\\
> p_n - 2 s_{1,n} - s_{2,n} + s_{1,n-1} + s_{2,n-1} \le 2 & n > 1\\
> \end{cases}.
> $$
>
> These constraints guarantee that at each bit column, the number of remaining signals after compression is feasible for a prefix-adder.
>
> **2. How $\mathbf{s}$ Interacts with the DAG and the Rest of the Framework**
>
> In ARITH-DAS, the allocation matrix $\mathbf{s} \in \mathbb{S}$ defines the architectural backbone of the compressor tree and **determines the search space for subsequent interconnection optimization**.
>
> Once a valid $\mathbf{s} \in \mathbb{S}$ is sampled by the Circuit Genetic Evolution (CGE) module, each compressor is assigned a stage number represented by $\mathcal{S}\in \mathbb{N} _{+}^{S\times T\times N}$, where:
>
> $$
> \sum_{i=1}^{S}\mathcal{S} _{i,t,n} =\mathbf{s} _{t,n}.
> $$
>
> The number of remaining partial products at stage $i$ and column $n$ is then updated as:
>
> $$
> \mathbf{P}_{i,n} = \begin{cases}
> p_n & i=1 \\\\
> \mathbf{P} _{i-1,n} - 2 \mathcal{S} _{i-1,1,n} - \mathcal{S} _{i-1,2,n} & i>1,n=1\\\\
> \mathbf{P} _{i-1,n} - 2 \mathcal{S} _{i-1,1,n} - \mathcal{S} _{i-1,2,n} + \mathcal{S} _{i-1,1,n-1} + \mathcal{S} _{i-1,2,n-1} & \text{otherwise}\\
> \end{cases}.
> $$
>
> $$
> \mathbf{P} _{i,n} \ge 0,
> $$
>
> where $\mathbf{P} _{i,n}\in \mathbb{N} _+$ is the remaining partial products at stage $i$.
>
> In our algorithm, we first greedily assign as many 3:2 Compressors as possible, and then assign 2:2 Compressors if needed. Since each compressor is associated with a specific stage number, **we enforce that compressors may only connect to compressors in the subsequent stages, thus forming a DAG (directed acyclic graph) search space**. This additional explanation will be added to Appendix C.3.2 in the revised version.
>
> The DAG derived from $\mathbf{s}$ enumerates the valid connection candidates among instantiated nodes. The interconnection optimization module then predicts fine-grained edges on top of this DAG under legality constraints (acyclicity, stage order, and port compatibility). Therefore, **$\mathbf{s}$ determines the node set and connection options, thus constraining the fine-grained search space of ARITH-DAS**.
>
> In summary, the allocation matrix $\mathbf{s}$ defines the coarse-grained search space (compressor allocation) and establishes the structural foundation for the fine-grained interconnection search. We will add this clarification in the revised version to make the explanation more intuitive.
>
> ---
>
> ### **Weakness 2: Missing Critical Reference**
> > **There is one paper: Reinforcement Learning for Combinatorial Optimization [1] which might be associated with this submission**
>
> We have carefully read the reference you provided and found it to be an excellent work that is **highly relevant to our study**. We have added a citation to this work in the **revised version of our paper**. Thank you again for bringing it to our attention!
>
> ---
>
> ### Reference
>
> [1] Darvariu, Victor-Alexandru, Stephen Hailes, and Mirco Musolesi. "Graph reinforcement learning for combinatorial optimization: A survey and unifying perspective." arXiv preprint arXiv:2404.06492 (2024).

---

> > ### Author Response · Authors · 2025-08-08
> > **We are looking forward to your feedback.**
> >
> > Dear Reviewer CFYW,
> >
> > We are writing as the authors of the paper "High-Performance Arithmetic Circuit Optimization via Differentiable Architecture Search" (ID: 25851). We sincerely thank you for your time and efforts during the rebuttal process. The deadline for the author-reviewer discussion period is approaching. We are looking forward to your feedback to understand if our responses have adequately addressed your concerns. If so, we would be grateful for your acknowledgment that the revisions have sufficiently clarified and strengthened our work. If not, please let us know your further concerns, and we will continue actively responding to your comments. We sincerely thank you once more for your insightful comments and kind support.
> >
> > Best,
> > Authors

---

> > > ### Comment · Reviewer_CFYW · 2025-08-08
> > > **Reply to authors**
> > >
> > > Sorry for my late response due to travel. I read the authors’ response and it clarified the details that I asked for. So I raised my rating.

---

> > > > ### Author Response · Authors · 2025-08-08
> > > > **Sincere Gratitude for Your Review**
> > > >
> > > > Dear Reviewer CFYW,
> > > >
> > > > Thank you very much for your positive feedback! We sincerely appreciate your thoughtful comments and are especially grateful for the updated rating. Your suggestions helped us clarify the presentation of our framework and have significantly strengthened the paper. Once again, thank you for your time and thoughtful review. We hope you had an enjoyable and refreshing trip!
> > > >
> > > > Best regards,
> > > > Authors

---

### Official Review · Reviewer_B7rW · 2025-07-06

**Clarity:** 3
**Significance:** 3
**Originality:** 3
**Rating:** 5
**Confidence:** 4

**Summary:**

This paper introduces a differentiable architecture search framework for arithmetic circuit optimization, which is set to overcome the existing limitations such as the under-explored design space of interconnection and the misaligned/oversimplified proxy objective, where both challenges are illustrated by case studies with randomized interconnect experiments. The empirical results show that the proposed ARITH-DAS framework can achieve 27.05% gain in hypervolume and 6.59% delay reduction in large-scale designs.

**Questions:**

N/A

**Ethical Concerns:**

["NO or VERY MINOR ethics concerns only"]

**Final Justification:**

I will keep my positive score (5). The authors have done important post submission experiments to clarify some very important statements. Although the back-end proxy has lowered the overall improvements, I think this paper is still valuable. However, I don't know if this is possible but we wanna to make sure the authors do include those back-end experiments and discussions if we accept it.

**Limitations:**

The proposed work is "proxy-free" in the sense of synthesis results. What about plugging into the full design flow? It looks like the proxy-gap is limited to a certain degree.

**Quality:**

3

**Strengths And Weaknesses:**

- This paper presents a comprehensive differentiable architecture search framework for arithmetic circuits, especially for the multipliers and multiply-accumulate units since they account for major computations.

- The paper is well written with easy-to-follow flows and illustrative figures to show the major contributions.

- The final results show a decent improvement of hypervolume on a suite of selected circuits, as well as significant delay reduction on large-scale designs.

However, I do spot certain weaknesses of the work and raise a few questions as follows:

- The authors mention in line 68, “The entire framework is trained in a proxy-free manner.” However, it seems that this applies to the post-synthesis alignment only.
- The major evaluation metric is hypervolume (HV), and the authors showcase a significant improvement on Pareto frontiers when compared to other baselines. However, with the large-scale computing circuits, the results are limited to the delay reduction only. What is the main reason that area (or more generally, Pareto frontiers) are not shown here?
- In the ablation study, it looks like neglecting certain key modules will not affect the improvements in general. Can the authors add the runtime difference when these key modules are not considered, since the current total runtime is not outperforming certain baselines?
- Certain typos are found. For example, line 116 should be “Misaligned”.

---

> ### Author Rebuttal · Authors · 2025-07-30
>
> ## **Response to Reviewer B7rW**
>
> We sincerely thank you for your careful review and for your constructive, insightful, and valuable comments. Please find below our detailed responses to your inquiries.
>
> ### **Weakness 1 & Limitation 1: Plugging into the Full Design Flow**
> > **The proposed work is "proxy-free" in the sense of synthesis results. What about plugging into the full design flow?**
>
> We sincerely thank you for your valuable suggestions. We have **integrated our framework, along with all baseline methods, into the complete design flow**, and present the corresponding results in the table below.
>
> |  | 16bit-AND | || 32bit-AND ||
> |:---:|:---:|:---:|:---:|:---:|:---:|
> |Methods| Post-Synthesis HV. | Full Flow HV. || Post-Synthesis HV. | Full Flow HV. |
> | Wallace  | $107.53$     | $27.49$    || $793.19$     | $83.95$      |
> | GOMIL    | $265.17_{+146.60\\%}$   | $109.91_{+299.82\\%}$ || $1216.48_{+53.37\\%}$   | $200.14_{+138.41\\%}$   |
> | UFO-MAC  | $269.18_{+150.33\\%}$   | $106.05_{+285.79\\%}$ || $1076.70_{+35.74\\%}$   | $159.57_{+90.08\\%}$    |
> | DOMAC    | $177.52_{+65.09\\%}$    | $53.41_{+94.31\\%}$   || $1267.22_{+59.76\\%}$   | $130.69_{+55.69\\%}$    |
> | ArithTreeRL | $315.19_{+193.12\\%}$   | $142.26_{+417.52\\%}$ || $2779.56_{+250.43\\%}$  | $898.79_{+970.67\\%}$   |
> | MUTE     | $310.89_{+189.12\\%}$   | $158.20_{+475.51\\%}$ || $2470.47_{+211.46\\%}$  | $663.50_{+690.39\\%}$   |
> ||
> | Ours     | $361.84_{+236.50\\%}$   | $163.26_{+493.90\\%}$ || $3247.06_{+309.37\\%}$  | $1079.97_{+1186.50\\%}$ |
>
> **1. Rationale for Using Post-Synthesis Metrics**
>
> Our framework can be **seamlessly integrated into the full design flow**, encompassing floorplanning, placement, clock tree synthesis, and routing. However, executing the complete flow is **computationally intensive**, and the resulting metrics are often **highly sensitive to hyperparameter** settings. Consequently, post-synthesis metrics have become the de facto standard in prior works [1,2,3], which we also adopt to ensure **a fair and consistent comparison**.
>
> **2. Integration into the Full Design Flow**
>
> We **integrate our framework into the full design flow** and summarize the results in the table above. Due to limited runtime and computation resources, we evaluate the full flow on 16-bit AND multipliers and 32-bit AND multipliers as representatives.
>
> Specifically, we build upon the `test/flow.tcl` script from the OpenROAD project to complete the EDA backend workflow, which includes `initialize_floorplan`, `tapcell`, `pdngen`, `place_pins`, `global_placement`, `detailed_placement`, `clock_tree_synthesis`, `global_route`, and other essential stages. During the flow, the floorplan is initialized with $0.88$ utilization as suggested by the OpenROAD tool, and `aspect_ratio` is set to $1.0$. The global placement density is set to $0.3$ for a faster evaluation, as the default in the `test/flow.tcl` script.
>
> **3. Discussion on the Proxy-Gap**
>
> **The longer and more complex the backend synthesis flow, the larger the proxy gap tends to be in proxy-based methods.** This is primarily due to the **intricate processing stages and optimization algorithms** in the backend, which make it difficult to accurately approximate final outcomes using proxy objectives at the frontend, especially when the results are **highly sensitive to backend parameters**.
>
> To further illustrate the impact of the proxy gap, we compute the average relative improvement of proxy-based and proxy-free methods with or without full design flow. Then we compare the relative gains before and after incorporating the full flow. As detailed in the table below, we observe that proxy-based methods **exhibit smaller relative gains after applying the full design flow**, highlighting the limitations of proxy objectives in capturing backend-accurate performance.
>
> ||16bit-AND||32bit-AND| |
> |:---:|:---:|:---:|:---:|:---:|
> ||Post-Synthesis Avg. HV.|Full-Flow Avg. HV.|Post-Synthesis Avg. HV.|Full-Flow Avg. HV.|
> | Wallace  | $107.53$     | $27.49$    | $793.19$     | $83.95$      |
> ||
> |Proxy-Based| $237.29_{+120.67\\%}$ | $89.79_{\substack{+226.62\\%\\\\ \color{orange}{\uparrow \times 1.87}}}$ | $1186.8_{+49.62\\%}$ | $163.46_{\substack{+94.71\\%\\\\ \color{orange}{\uparrow \times 1.91}}}$ |
> |Proxy-Free| $329.31_{+206.25\\%}$ | $154.57_{\substack{+462.27\\%\\\\ \color{orange}{\uparrow \times 2.24}}}$ | $2832.36_{+257.08\\%}$ | $880.75_{\substack{+949.13\\%\\\\ \color{orange}{\uparrow \times 3.69}}}$ |
>
> ---
>
> ### **Weakness 2: Missing Large-Scale Computing Circuits Pareto/area Results**
> > **However, with the large-scale computing circuits, the results are limited to the delay reduction only. What is the main reason that area (or more generally, Pareto frontiers) are not shown here?**
>
> **1. On the Omission of Area Metrics in the Main Paper**
>
> **First**, due to **space constraints**, we report the area metrics of the large-scale computing circuit in Table 10 of Appendix D, as our primary focus lies in interconnection optimization, where **delay is the most relevant performance metric**.
> **Second**, a complete computing system comprises not only arithmetic circuits but also control logic, I/O interfaces, memory blocks, and other components. Consequently, **area optimization within arithmetic submodules has a limited impact on the overall area**. For this reason, we present area-related results in the appendix rather than in the main text.
>
> **2. On the Absence of Full Pareto Frontier Results**
>
> **First**, evaluating complete Pareto frontiers requires extensive experimentation, but logic synthesis and simulation of large-scale computing circuits are **extremely time-consuming and computationally expensive**. To balance feasibility and thoroughness, we conduct a full Pareto frontier evaluation only on a commonly shared computational structure, **the Processing Element (PE) array**, following previous works [1, 2]. The corresponding results are presented in Figure 7 of Section 5.3.
>
> **Second**, as noted earlier, large-scale computing circuits are **composed of multiple heterogeneous substructures**. Area optimization on arithmetic circuits alone often **yields limited overall area improvement**. Therefore, we focus on the delay metric and omit the full Pareto frontier analysis for the entire large-scale design.
>
> ---
>
> ### **Weakness 3: Runtime for Ablation Study**
> > **Can the authors add the runtime difference when these key modules are not considered, since the current total runtime is not outperforming certain baselines?**
>
> Thanks for your suggestion. We conclude **the runtime for the ablation study**, where several key modules are removed, in the following table. We will add those results in the revised version of our submission.
>
> | | 16-bit AND |  | 32-bit AND |  |
> |:---:|:---:|:---:|:---:|:---:|
> |  | Runtime (hours) $\downarrow$  | HV. $\uparrow$  | Runtime (hours) $\downarrow$  | HV.$\uparrow$ |
> | **Ours** | $13.06$ | $361.84$ | $38.73$ | $3247.06$  |
> ||
> | **w/o CGE** | $10.28_{-21.29\\%}$ | $326.58_{-9.74\\%}$ | $27.92_{-27.91\\%}$ | $2427.64_{-25.24\\%}$ |
> | **w/o MRG** | $11.10_{-15.01\\%}$ | $313.57_{-13.34\\%}$ | $26.32_{-32.04\\%}$ |  $2499.68_{-23.02\\%}$ |
> | **w/o PPO** | $9.18_{-29.71\\%}$  | $326.49_{-9.77\\%}$ | $22.37_{-42.24\\%}$ | $3007.14_{-7.39\\%}$ |
>
> ---
>
> ### **Weakness 4: Typos**
> > **Certain typos are found.**
>
> We appreciate your careful reading and helpful feedback. The typo has been corrected. We have also conducted a thorough proofreading to eliminate similar minor errors throughout the paper.
>
> ---
>
> ### Reference
>
> [1] Zuo D, Ouyang Y, Ma Y. Rl-mul: Multiplier design optimization with deep reinforcement learning[C]//2023 60th ACM/IEEE Design Automation Conference (DAC). IEEE, 2023: 1-6.
>
> [2] Wang Z, Wang J, Zuo D, et al. A hierarchical adaptive multi-task reinforcement learning framework for multiplier circuit design[C]//Forty-first International Conference on Machine Learning. 2024.
>
> [3] Xiao W, Qian W, Liu W. GOMIL: Global optimization of multiplier by integer linear programming[C]//2021 Design, Automation & Test in Europe Conference & Exhibition (DATE). IEEE, 2021: 374-379.

---

> > ### Comment · Reviewer_B7rW · 2025-08-04
> > **Regarding the flow settings**
> >
> > The authors' response regarding the flow settings remains unconvincing and raises additional concerns. Recent industrial efforts—such as those by NVIDIA Research and the Synopsys Datapath/Synthesis team—have demonstrated the importance of full-flow evaluations, including backend closure. This is particularly critical given that the targeted design cases in this paper are considered small-scale in the context of modern digital processor pipelines. While the cited works [1–3] are valuable contributions, relying solely on front-end metrics without backend validation may present an incomplete or potentially misleading picture for future research. To ensure the robustness and practical relevance of the proposed approach, I strongly encourage the authors to include or discuss backend evaluation results, especially for the designs under consideration.

---

> > > ### Author Response · Authors · 2025-08-08
> > > **Response to Reviewer B7rW on Full-Flow Evaluation**
> > >
> > > We sincerely thank the reviewer for the insightful comments. We fully recognize the importance of full-flow evaluation as highlighted by recent industrial practices. In response, **we have performed comprehensive full-flow experiments** using the `OpenROAD-flow-scripts` framework [4] with `make do-finish` command, covering the complete flow from RTL Verilog to GDSII layout. The results indicate that **our method achieves the leading TNS and WNS in all evaluated cases**, demonstrating superior timing closure and backend convergence. **These results are now included in the revised manuscript, along with a dedicated analysis section**. We believe this strengthens the robustness and practical relevance of our manuscript.
> > >
> > > **1. Experiment Setup**
> > >
> > > To ensure fairness and end-to-end validity, we directly apply the multiplier designs generated by our method and all baselines to the full backend flow.
> > > We evaluate:
> > >
> > > - Representative circuit: 16-bit and 32-bit AND multipliers (Tables 1–2).
> > > - Large-scale integration: Transformer attention layer and PE array (Tables 3–4).
> > >
> > > We report key backend metrics: **TNS**, **WNS**, **instance area** and **total power**, including those **not directly** optimized, to provide a holistic view.
> > >
> > > **2. Results Summary**
> > >
> > > Across both evaluations, **our method consistently achieves the best TNS and leading WNS, two key indicators of backend timing quality [5, 6], suggesting strong robustness in timing closure**. While it may not always yield the lowest area or power, our approach achieves **a well-balanced trade-off across all key design metrics**. In large-scale integration scenarios, our method further sustains excellent timing performance while maintaining competitive area and power efficiency, confirming its **practical scalability and robustness in full-flow backend design**.
> > >
> > > > Table 1. 16bit-AND multiplier
> > >
> > > | Method | tns ($ns$) $\uparrow$ | wns ($ns$) $\uparrow$ | instance area ($\mu m^2$) $\downarrow$ | power ($n W$) $\downarrow$ |
> > > |:---:|:---:|:---:|:---:|:---:|
> > > | Wallace | -11.1723 | -0.51446 | 4026.97 | 0.0220279 |
> > > | GOMIL | -12.829 | -0.584538 | 3853.28 | 0.0209472 |
> > > | UFO-MAC | -10.0185 | **-0.44502** | **3082.67** | 0.0181502 |
> > > | DOMAC | -13.14 | -0.601615 | 3784.38 | 0.021324 |
> > > | ArithTreeRL | -11.2985 | -0.516607 | 3429.01 | 0.0172025 |
> > > | MUTE | -10.7299 | -0.496157 | 3253.98 | **0.0147275** |
> > > | Ours | **-9.88311** | -0.471682 | 3867.11 | 0.0205214 |
> > >
> > > > Table 2. 32bit-AND multiplier
> > >
> > > | Method | tns ($ns$) $\uparrow$ | wns ($ns$) $\uparrow$ | instance area ($\mu m^2$) $\downarrow$ | power ($n W$) $\downarrow$ |
> > > |:---:|:---:|:---:|:---:|:---:|
> > > | Wallace | -37.2904 | -0.792821 | 10695.9 | 0.101978 |
> > > | GOMIL | -34.4961 | -0.780473 | 10734.4 | 0.103065 |
> > > | UFO-MAC | -33.6154 | -0.741069 | 10507.5 | **0.0893294** |
> > > | DOMAC | -36.193 | -0.77785 | 11293 | 0.101894 |
> > > | ArithTreeRL | -32.9907 | -0.746786 | 10548.5 | 0.107055 |
> > > | MUTE | -34.9426 | -0.747897 | **10475.6** | 0.0985021 |
> > > | Ours | **-32.7633** | **-0.727951** | 10906.5 | 0.105757 |
> > >
> > > > Table 3. Attention layer
> > >
> > > | Method | tns ($ns$) $\uparrow$ | wns ($ns$) $\uparrow$ | instance area ($\mu m^2$) $\downarrow$ | power ($n W$) $\downarrow$ |
> > > |:---:|:---:|:---:|:---:|:---:|
> > > | Wallace | -1417.44 | -0.879366 | 1.40104e+06 | 7.35242 |
> > > | GOMIL | -870.798 | -0.673238 | 1.40468e+06 | 6.62554 |
> > > | UFO-MAC | -907.867 | -0.669357 | 1.39698e+06 | **6.25145** |
> > > | DOMAC | -1064.7 | -0.781636 | 1.40022e+06 | 8.08283 |
> > > | ArithTreeRL | -602.81 | -0.621154 | 1.39229e+06 | 7.49223 |
> > > | MUTE | -1281.07 | -0.594251 | **1.39048e+06** | 7.22216 |
> > > | Ours | **-585.747** | **-0.573738** | 1.39775e+06 | 6.94129 |
> > >
> > > > Table 4. Process element array (PE array)
> > >
> > > | Method | tns ($ns$) $\uparrow$ | wns ($ns$) $\uparrow$ | instance area ($\mu m^2$) $\downarrow$ | power ($n W$) $\downarrow$ |
> > > |:---:|:---:|:---:|:---:|:---:|
> > > | Wallace | -1091.16 | -0.497151 | 874830 | 0.436699 |
> > > | GOMIL | -867.903 | -0.468943 | 865828 | 0.434043 |
> > > | UFO-MAC | -824.201 | -0.468148 | **825849** | 0.472914 |
> > > | DOMAC | -772.635 | -0.447722 | 883844 | 0.479212 |
> > > | ArithTreeRL | -645.562 | -0.474561 | 833118 | **0.433875** |
> > > | MUTE | -869.457 | **-0.350233** | 852462 | 0.474226 |
> > > | Ours | **-607.886** | -0.421839 | 838346 | 0.484758 |
> > >
> > >
> > > ## Reference
> > >
> > > [4] Ajayi T, Chhabria V A, Fogaça M, et al. Toward an open-source digital flow: First learnings from the openroad project[C]//Proceedings of the 56th Annual Design Automation Conference 2019. 2019: 1-4.
> > >
> > > [5] Kahng A B, Lienig J, Markov I L, et al. VLSI physical design: from graph partitioning to timing closure[M]. Netherlands: Springer, 2011.
> > >
> > > [6] Geng Z, Wang J, Liu Z, et al. LaMPlace: Learning to optimize cross-stage metrics in macro placement[C]//The Thirteenth International Conference on Learning Representations.

---

> > > > ### Comment · Reviewer_B7rW · 2025-08-08
> > > >
> > > > Thanks for the OpenROAD results.
> > > >
> > > > Can you clarify 1) which technology library is used? 2) what is the timing target and are the physical constraints the same for all the designs and 3) was the synthesis and physical design configurations delay-driven or area-driven ?  In general the results are very sensitive to the timing target and also the synthesis stage. It looks to me the setups have a very relaxed timing constraint with a relative fast technology node.
> > > >
> > > > Nevertheless, the paper has proposed interesting ideas and overall is positive. But the results into deeper design stacks indeed show degradation of improvements. To be honest, the follow-up rebuttal experiments confirm my assumption - In a more strong EDA tool setups (e.g., high effort synthesis and p&r) or commercial tools, I assume the results will not be very visible. It is good for the authors to really clarify the advantages and disadvantages of the work to avoid potential misleading message for others in the future.

---

> > > > > ### Author Response · Authors · 2025-08-08
> > > > > **Response to Reviewer B7rW**
> > > > >
> > > > > We sincerely thank the reviewer for the valuable follow-up comments and thoughtful reflections on our additional experiments. Please find our clarifications below:
> > > > >
> > > > > **1. Technology Library:**
> > > > > We used the **Nangate45** open-source standard-cell library for all experiments in accordance with our submission, which is widely adopted in academic EDA research and ensures consistency across designs.
> > > > >
> > > > > **2. Timing Target and Physical Constraints:**
> > > > > For the evaluations on **16-bit AND and 32-bit AND multipliers**, we set the clock period to **0.5ns**. For the **attention layer and processing element array**, we used a **1.2ns** clock period. All designs were implemented with **identical physical constraints** to ensure a fair comparison across all methods.
> > > > >
> > > > > **3. Synthesis and Physical Design Configuration (Area vs Delay Driven):**
> > > > > We used an **area-driven** configuration. Specifically, our flow is based on the reference setup in `OpenROAD-flow-scripts/flow/designs/nangate45/gcd/config.mk`, which is one of the most representative configurations. We have included our `config.mk` and `constraint.sdc` files below for transparency and reproducibility.
> > > > >
> > > > > As the reviewer insightfully noted, the distinction between post-synthesis evaluation and full backend validation is indeed critical. **We will include a dedicated discussion in the revised submission to explicitly acknowledge this limitation.**.
> > > > >
> > > > > Moreover, this perspective highlights a promising direction for future work: **enabling training and adaptation of our framework within full-design flow environments, thereby enhancing its practical applicability to industry-grade pipelines**. To make such integration feasible, it may be necessary to incorporate fast, parameter-adaptive evaluation techniques, such as surrogate environment models or learned cost estimators, to address the high computational overhead of full-flow iterations.
> > > > >
> > > > > We are grateful for your insightful comments, which sharpen the scope of our contribution and guide its extension to real-world deployment.
> > > > >
> > > > > ```makefile
> > > > > # config.mk
> > > > > export BENCH_NAME ?= <benchmark name>
> > > > > export METHOD_NAME ?= <method name>
> > > > > export MUL_FILE ?= <path/to/multiplier_design.v>
> > > > > export BENCH_FILE ?= <path/to/benchmark_design.v>
> > > > > export BUILD_DIR ?= <path/to/build_dir>
> > > > >
> > > > > export DESIGN_NICKNAME ?= ${BENCH_NAME}_${METHOD_NAME}
> > > > > export DESIGN_NAME ?= top
> > > > > export PLATFORM    = nangate45
> > > > >
> > > > > export VERILOG_FILES = ${BENCH_FILE} ${MUL_FILE}
> > > > >
> > > > > export WORK_HOME=$(BUILD_DIR)/${BENCH_NAME}/${METHOD_NAME}
> > > > >
> > > > > export SDC_FILE      ?= <path/to/constraint.sdc>
> > > > > export ABC_AREA      = 1
> > > > >
> > > > > export CORE_UTILIZATION ?= 55
> > > > > export PLACE_DENSITY_LB_ADDON = 0.20
> > > > > export TNS_END_PERCENT        = 100
> > > > > export REMOVE_CELLS_FOR_EQY   = TAPCELL*
> > > > > ```
> > > > >
> > > > > ```sdc
> > > > > # constraint.sdc
> > > > > current_design top
> > > > >
> > > > > set clk_name clk
> > > > > set clk_port_name clk
> > > > > set clk_period <period>
> > > > > set clk_io_pct 0.2
> > > > >
> > > > > set clk_port [get_ports $clk_port_name]
> > > > >
> > > > > create_clock -name $clk_name -period $clk_period $clk_port
> > > > >
> > > > > set non_clock_inputs [all_inputs -no_clocks]
> > > > >
> > > > > set_input_delay [expr $clk_period * $clk_io_pct] -clock $clk_name $non_clock_inputs
> > > > > set_output_delay [expr $clk_period * $clk_io_pct] -clock $clk_name [all_outputs]
> > > > >
> > > > > ```

---

### Note · Authors · 2025-08-13

Dear Area Chair and Reviewers,

We are grateful for your time and expert feedback. For your convenience, we have prepared a summary of our responses and outlined how we have addressed the reviewers' concerns as follows.

The reviewers highlighted the following key strengths of our paper:
- **Clarity and Rigor** [Reviewer B7rW, Reviewer CFYW, Reviewer tJhd] Commended for its clear presentation, supported by illustrative figures and mathematical formalism.
- **Novelty and Significance** [Reviewer B7rW, Reviewer tJhd] A novel framework that innovatively addresses the unexplored problem of arithmetic circuit interconnections.
- **Effectiveness and Performance** [Reviewer B7rW, Reviewer CFYW, Reviewer tJhd] Demonstrates high effectiveness, with ablation studies confirming validity and achieving state-of-the-art results.

We have diligently worked to address each of the reviewers' concerns in detail and outline how we have addressed the concerns raised by each reviewer as follows.

## Reviewer B7rW
> **1. Full-Flow Evaluation** We integrated our framework into a complete EDA full-flow, demonstrating superior timing closure and backend convergence with leading TNS and WNS metrics.

> **2. Clarification on Large-Scale Evaluation** We explained the absence of full Pareto frontiers for large-scale designs due to computational cost.

> **3. Runtime for Ablation Study** We have concluded the runtime for the ablation study where several key modules are removed.

> **4. Editorial Improvements** We have conducted a thorough proofreading to eliminate typo errors throughout the paper.

> **5. Comprehensive Limitation Discussion** We included a new discussion on the limitations of post-synthesis metrics and outlined promising future research directions.

## Reviewer CFYW
> **6. Unclear Description of the Method** We provided further demonstration of the compressor allocation matrix to clarify its definition, role, and how it interacts with the rest of the framework.

> **7. Missing Critical Reference** We have added a citation of a highly relevant work in our paper.

## Reviewer tJhd
> **8. Integrating with Prior Works** We have embedded our method into two prior frameworks, showing consistent performance improvements and validating its modularity.

> **9. Performance Analysis with MUTE** We addressed the area comparison with MUTE, attributing our superior results to a delay-aware generator that optimizes the area-delay trade-off.

Best regards,

The Authors

---

### Decision · Program_Chairs · 2025-09-17

**Decision:**

Accept (spotlight)

**Comment:**

The final scores for the paper unanimously recommend acceptance. Although 2 out of 3 of them come with relatively low confidence, reviewer B7rW - after engaging in a very detailed discussion during the rebuttal - very explicitly voices his support for the paper with enough confidence to (in my opinion) diminish the uncertainty arising from the other reviews.

 For the sake of completeness, all reviewers agree that the proposed method is novel and practical, evaluation shows promising results, and the paper is clearly written and well-argued. As far as shortcomings go: all reviewers asked for clarifications with respect to some aspects of the paper and/or other works, all of which the authors were able to address. From the discussion with reviewer B7rW: the most important missing bit, that was added during the rebuttal, was closure timing experiments on the OpenROAD dataset, which the authors say have been incorporated into a revised paper. The results show diminishing returns in a complex setting, but not to the point where it would undermine contributions of the paper - I would urge the authors to accurately comment on these results and potential arising limitations.

Overall, I find sufficient reasons to recommend acceptance. In particular, I think the paper stands out due to its technical depth in combining two seemingly unrelated fields of differentiable search and circuit optimisation. Together with the in-depth evaluation covering both post-synthesis and full-flow scenarios (the latter coming from the rebuttal), I would consider it to be a particularly noticeable effort in exploring and characterising new directions. Whether this direction will stand the test of time, time will tell, but the submitted work deserves recognition, in my opinion.